# Uncovering new families and folds in the natural protein universe

Janani Durairaj[1,2], Andrew M. Waterhouse[1,2], Toomas Mets[3,4], Tetiana Brodiazhenko[3], Minhal Abdullah[3,4], Gabriel Studer[1,2], Gerardo Tauriello[1,2], Mehmet Akdel[5], Antonina Andreeva[6], Alex Bateman[6], Tanel Tenson[3], Vasili Hauryliuk[3,4,7,8], Torsten Schwede[1,2 ✉] & Joana Pereira[1,2 ✉]

We are now entering a new era in protein sequence and structure annotation, with hundreds of millions of predicted protein structures made available through the AlphaFold database[1]. These models cover nearly all proteins that are known, including those challenging to annotate for function or putative biological role using standard homology-based approaches. In this study, we examine the extent to which the AlphaFold database has structurally illuminated this 'dark matter' of the natural protein universe at high predicted accuracy. We further describe the protein diversity that these models cover as an annotated interactive sequence similarity network, accessible at https://uniprot3d.org/atlas/AFDB90v4. By searching for novelties from sequence, structure and semantic perspectives, we uncovered the β-flower fold, added several protein families to Pfam database[2] and experimentally demonstrated that one of these belongs to a new superfamily of translation-targeting toxin–antitoxin systems, TumE–TumA. This work underscores the value of large-scale efforts in identifying, annotating and prioritizing new protein families. By leveraging the recent deep learning revolution in protein bioinformatics, we can now shed light into uncharted areas of the protein universe at an unprecedented scale, paving the way to innovations in life sciences and biotechnology.

Since the sequencing of the first protein, large-scale efforts brought about by faster and cheaper genome sequencing techniques have shed light into some of the sequences that nature has sampled so far. At present, there are more than 350 million unique protein coding sequences deposited in UniProt and more than 3 billion in MGnify[3,4]. The rate at which these data are growing is much faster than experimental functional characterization. To close the gap, functional information is gathered for a subset of proteins and the findings extrapolated to close homologues. Manual curation is carried out by those assembling the genomes and by biocurators[5] and incorporated into automated annotation pipelines such as InterPro[6].

Despite the great success of such approaches, only 83% of UniProt sequences are covered by InterPro, and many correspond to domains of unknown function (DUF). Thus, numerous protein sequences remain functionally unannotated and unclassified. Some of these may just correspond to divergent forms of known protein families that lie beyond the detection horizon of automated, homology-based methods; others could belong to so-far undescribed protein families with yet-to-be determined molecular or biological functions[7].

The three-dimensional (3D) structure of a protein is intrinsically linked with its molecular function. Experimental structure determination is an expensive and time-consuming process, and homology-based computational prediction loses its power for proteins without close homologues[8]. Notwithstanding, deep learning-based approaches have recently achieved unprecedented accuracy, with AlphaFold2 at the forefront. Its success drove the establishment of the AlphaFold database (AFDB), which contains predicted structural models for about 215 million natural protein sequences from UniProt, including many of the unannotated proteins. At the same time, deep learning-based approaches have also recently been used for predicting functional properties from structure[9] and protein names from sequence[10].

In this work, we combine sequence similarities and structure features with deep learning-based function prediction tools to shed light on 'functionally dark' proteins in UniProt. We revised their proportion, evaluated how many of them now have high-confidence structural models that can be leveraged for downstream analysis, and constructed an annotated and interactive sequence similarity network with millions of proteins. By exploring this network, we discovered 290 putative new protein families, identified at least one new protein fold and defined a new superfamily of translation-targeting toxin–antitoxin (TA) systems that we experimentally validated and dubbed TumE–TumA. This work demonstrates that functional annotation of proteins, even from a purely computational perspective, requires a combination of data sources and approaches, which become increasingly available and attainable due to the rapid and continuing advances at the interface between life sciences and deep learning.

[1]Biozentrum, University of Basel, Basel, Switzerland. [2]SIB Swiss Institute of Bioinformatics, University of Basel, Basel, Switzerland. [3]Institute of Technology, University of Tartu, Tartu, Estonia. [4]Department of Experimental Medical Science, Lund University, Lund, Sweden. [5]VantAI, New York, NY, USA. [6]European Molecular Biology Laboratory, European Bioinformatics Institute (EMBL-EBI), Hinxton, UK. [7]Science for Life Laboratory, Lund, Sweden. [8]Virus Centre, Lund University, Lund, Sweden. ✉e-mail: torsten.schwede@unibas.ch; joana.pereira@unibas.ch

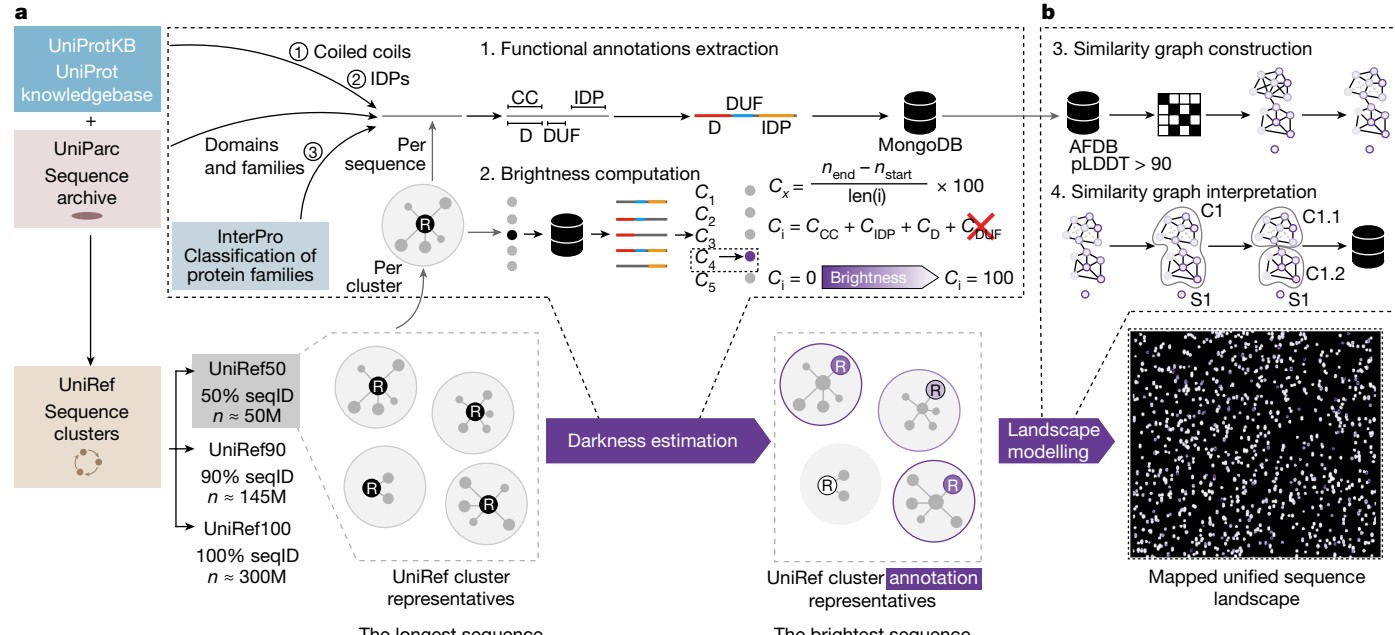

**Fig. 1 | General workflow for the collection, classification and mapping of functionally dark proteins in UniProt and AFDB. a**, Starting from the clusters in UniRef50, we collected all the functional annotations for all included UniProtKB and UniParc entries, including domain (D), coiled-coil (CC) and intrinsically disordered (IDPs) predictions and excluding all of those with putative, hypothetical, uncharacterized and DUF in their names. $C_x$ corresponds to the coverage of an annotation, $C_i$ corresponds to the functional brightness

across the entire sequence. We selected the protein with the highest full-length annotation coverage (that is, brightness, $C_i$) as the functional representative of each cluster. **b**, From the collected UniRef50 clusters, we selected those with a structural representative with pLDDT greater than 90 in the AFDB v.4, and constructed a large-scale sequence similarity network by all-against-all MMseqs2 searches, representing the sequence landscape of more than 6 million UniRef50 clusters.

## Functional darkness in UniProt and AFDB

As of August 2022, there were more than 350 million unique protein sequences in UniProt (that is, UniRef100 clusters[11]). We focus our analysis on these as they have a higher confidence than those deposited in metagenomics databases such as MGnify. These sequences correspond to roughly 50 million non-redundant proteins when clustered to a maximum sequence identity of 50% (UniRef50). Starting from these clusters, we define the 'functional brightness' of a given protein as the full-length coverage with annotations of its close homologues, and a UniRef50 cluster is as 'bright' as the 'brightest' sequence it encompasses (Fig. 1a). For that, we only considered those annotations that correspond to domains and families whose title does not include 'putative', 'hypothetical', 'uncharacterized' and 'DUF', but included predicted coiled-coil and intrinsically disordered segments to focus our analysis solely on functionally dark proteins with a potential for a globular (or other) fold type.

We found that 34% of all UniRef50 clusters (10% of UniRef100, roughly 34 million unique proteins) are dark as they do not reach a functional brightness higher than 5% (Extended Data Fig. 1a). Whereas the brightness of a cluster is not directly proportional to the number of sequences within it (Pearson correlation coefficient of zero), bright clusters (functional brightness greater than or equal to 95%) tend to be larger than those whose members are poorly annotated (mean $19 \pm 123$ unique sequences in bright clusters compared to $2 \pm 7$ in dark).

Whereas UniRef50 clusters encompass sequences from the UniProt Knowledgebase (UniProtKB) and the UniProt Archive (UniParc)[12], the latest version of AFDB (v.4) covers only UniProtKB and excludes both long and viral sequences. Consequently, 78% of all UniRef50 clusters have members with a predicted structure in AFDB (Extended Data Fig. 1b). Of these, 29% are functionally dark, a proportion that drops with an increase in predicted model accuracy (Extended Data Fig. 1c,d) while retaining a similar proportion of DUFs (Extended Data Fig. 1e).

Thus, there is a considerable proportion of proteins in UniProt that cannot be automatically annotated, but that high-confidence structural information can now be leveraged to gain insights about many of these.

## Sequence similarity network of AFDB90

Whereas UniRef50 provides groups of sequences that are overall similar at the sequence level, they do not reach the family and superfamily levels and do not account for local similarities. To reach these levels and put functionally dark clusters into evolutionary context, we constructed a large-scale sequence similarity network of all clusters where structural information can be confidently leveraged to support functional annotations. This corresponds to the 6,136,321 UniRef50 clusters (roughly 53 million unique protein sequences) that have structural representatives with an average predicted local distance difference test (pLDDT) score more than 90 in AFDB (the AFDB90 dataset).

We used MMseqs2 (ref. 13) for all-against-all sequence searches (Fig. 1b), connecting two sequences if they have an alignment that covers at least 50% of one of the proteins with an $E$ value $< 1 \times 10^{-4}$. The resulting network has more than 4 million connected nodes and 10 million edges, which includes 43% of all dark UniRef50 clusters (Fig. 2). Of these dark clusters, 40% connect to bright UniRef50 clusters, revealing potential evolutionary relationships for more than 700,000 unique proteins.

The network is composed of 242,876 connected components with at least two nodes, with the largest encompassing about 50% of all AFDB90 (Fig. 2a). Of these components, 19% have an average brightness content below 5% ('fully dark') (Fig. 2d). Only 25% of the components are 'fully bright' (that is, average functional brightness more than 95%). The percentage of UniRef50 clusters in fully dark components decreases with the component's size (Fig. 2b,c), highlighting that the lower the number of homologues the harder a protein is to annotate. Still, and while the distribution is skewed towards smaller sizes in both fully dark

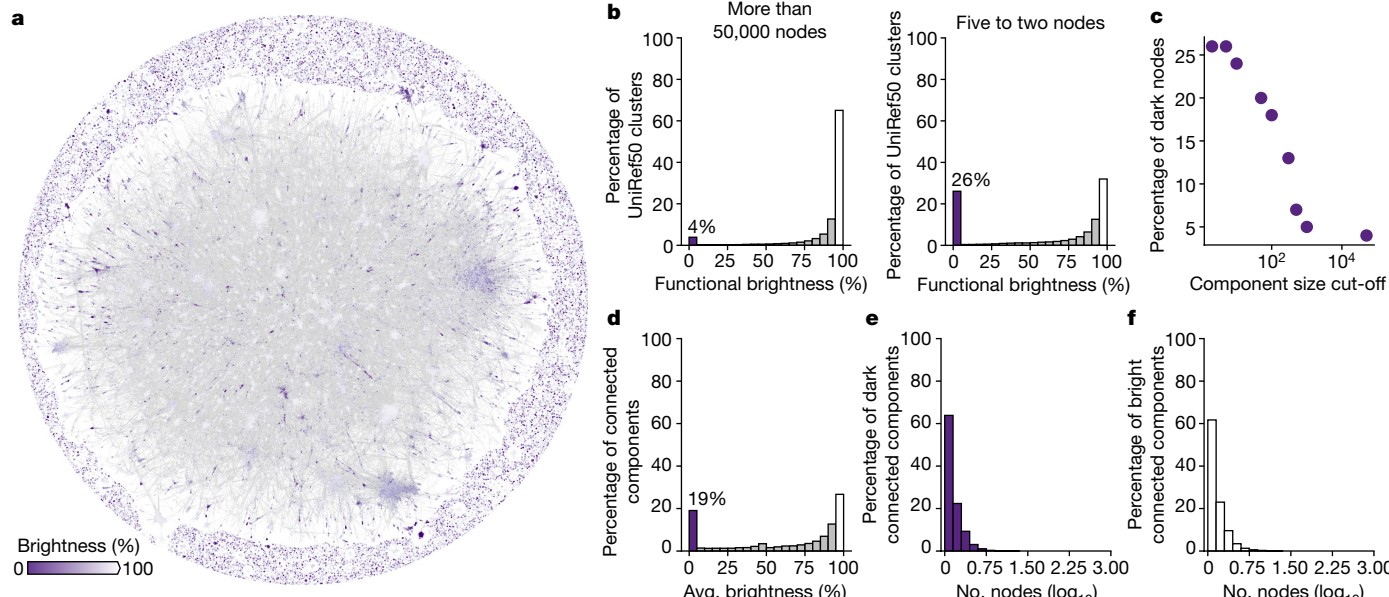

**Fig. 2 | Large-scale sequence similarity network for more than 6 million UniRef50 cluster representatives with high predicted accuracy models in AFDB (AFDB90). a**, Layout of the resulting network, as computed with Cosmograph (https://cosmograph.app/). The network contained 4,270,404 nodes connected by 10,339,158 edges, reduced for simplicity to a set of 688,852 communities connected by a total of 1,488,764 edges (see Methods section 'Large-scale sequence similarity network' for details). The 1,865,917 UniRef50 clusters that did not connect to any other in the MMseqs2 searches were excluded. Only the 473,612 communities that have at least one inbound or outbound edge (degree of 1) are shown in the figure. Nodes are coloured by the average functional brightness of the UniRef50 clusters included in the corresponding community. An interactive version is available at https://uniprot3d.org/atlas/AFDB90v4. **b**, Histograms of functional brightness content for connected components with more than 50,000 and with only five to two nodes (UniRef50 clusters), highlighting their different darkness content. **c**, Scatter plot of the component size (that is, number of UniRef50 clusters) cut-off and the percentage of functionally dark UniRef50 clusters. **d**, Histogram of the average (avg.) brightness per connected component. **e,f**, Size distribution for fully dark connected components (**e**, average brightness less than 5%) and fully bright connected components (**f**, average brightness more than 95%).

and fully bright components (Fig. 2e,f), the largest dark component in our network has more than 800 nodes. These fully dark components are fertile ground for new family discovery, as exemplified by the two new families we describe below.

## A new glycosyltransferase family

The largest functionally dark connected component in our set is component 27, with 836 UniRef50 clusters (4,889 unique bacterial protein sequences, average brightness 2 ± 13%, Fig. 3a). Their representatives have a median length of 665 ± 169 amino acids, most are predicted to be transmembrane and they are annotated as uncharacterized YfhO in InterPro. Indeed, the proteins in this component that are not called an uncharacterized protein mostly have the title YfhO family protein, which corresponds to a family involved in lipoteichoic acid or wall teichoic acid glycosylation[14]. However, the predicted structural model superposes poorly to the YfhO family (template modelling (TM) score 0.58, Fig. 3b), prompting a more in-depth investigation.

HHPred[15] and Foldseek[16] find many medium-to-high confidence matches in the Protein Data Bank (PDB) (probability more than 95% and TM score roughly 0.6, Fig. 3b), including the eukaryotic dolichyl-diphosphooligosaccharide-protein glycosyltransferase subunit STT3 and its bacterial homologue oligosaccharyltransferase PglB[17,18], absent from our network because their representatives have an average pLDDT less than 90. We collected sequences for all four groups of proteins (YfhO, STT3, PglB and component 27) and built a sequence similarity network to investigate how they may relate at the sequence level (Fig. 3a). This network highlighted that most dark proteins in component 27 cluster separately from the reference YfhO, forming a single YfhO-like protein family that is linked to the STT3/PglB groups by several hypothetical proteins, mostly of prokaryotic origin, often annotated as 'glycosyltransferase family 39 protein'.

These results support the notion that component 27 belongs to the well-studied superfamily of transmembrane oligosaccharyl- and glycosyltransferases, but also indicate that it is a hitherto undescribed bacterial protein family. In this case, inspecting the AlphaFold model revealed possible inconsistencies in their automated annotation, illustrating the added value of structural models to guide sequence-based family classification.

## A new TA superfamily

Component 159 is composed of 327 UniRef50 clusters, corresponding to 1,222 unique protein sequences, mostly annotated as DUF6516 (Fig. 4). These proteins are predicted to adopt a conserved α + β fold, where two α-helices pack against an antiparallel β-sheet with seven strands (Extended Data Fig. 2). Contrary to component 27, HHPred and Foldseek searches found no confident matches in the PDB. A high-resolution similarity network unravelled seven distinct classes of DUF6516-containing proteins (Fig. 4a).

On the basis of the AFDB models, structure-based function predictor DeepFRI[9] proposed that they may bind DNA or other nucleic acids and carry a hypothetical catalytic site with a hydrolase activity over ester bonds (Fig. 4c and Supplementary Table 1). Genomic context analysis with GCsnap[19] highlighted that DUF6516-coding genes are commonly found in a conserved two-gene (bicistronic) genomic arrangement, with DUF6516 predominantly located downstream of the conserved bicistronic 'partner' (clusters 1, 2, 4 and 6).

Whereas most of the partner genes associated with DUF6516 code for hypothetical proteins of unknown function, one in cluster 1 is a remote homologue of RelB, a well-characterized antitoxin[20]. Indeed, the bicistronic arrangement is typical for TA systems[21]. When active, the TA toxin proteins abolish bacterial growth, and the control of this toxicity is executed by the antitoxin, which, in the case of type II TA systems, is a protein that acts by forming an inactive complex with the

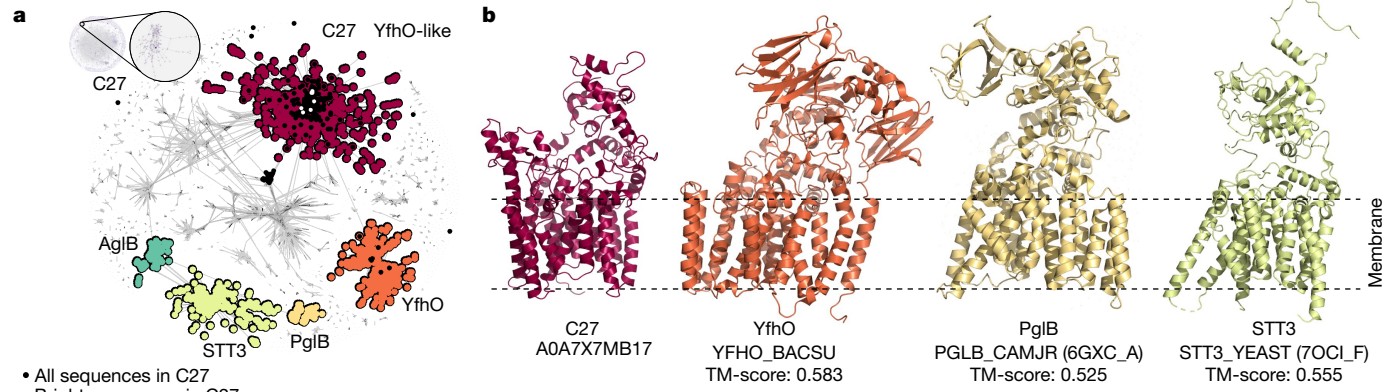

**a**

C27 YfhO-like

C27

AglB

STT3  PglB  YfhO

• All sequences in C27
○ Bright sequences in C27

**b**

C27
A0A7X7MB17

YfhO
YFHO_BACSU
TM-score: 0.583

PglB
PGLB_CAMJR (6GXC_A)
TM-score: 0.525

STT3
STT3_YEAST (7OCI_F)
TM-score: 0.555

Membrane

**Fig. 3 | Connected component 27 is a new family in a well-studied superfamily of transmembrane glycosyltransferases. a**, High-resolution sequence similarity network for 7,004 homologues of the sequences in component 27, computed with CLANS at an $E$ value threshold of $1 \times 10^{-20}$. Points represent individual proteins and grey lines BLASTp matches at an $E$ value $< 1 \times 10^{-20}$. Individual clusters are coloured and labelled according to their representative members. Only YfhO-like and STT3/PglB sequences are highlighted, with grey dots depicting other homologous groups. AglB corresponds to the PglB/STT3-like sequences from archaea. Black dots depict those sequences that make component 27 in our network, and white dots mark those that are bright. **b**, Predicted structural models as in AFDBv4 for the representative of component 27 (C27, UniProt ID A0A7X7MB17) and YfhO (UniProt ID YFHO_BACSU), and experimental structures of the PglB (PDB ID 6GXC, chain A) and STT3 (PDB ID 7OCI, chain F) cluster representatives. Models are coloured according to their corresponding cluster in **a**. The membrane regions, as predicted with the PPM v.3.0 server[39], are marked by dashed lines.

toxin. DeepFRI predictions for DUF6516 partners suggests they may also bind DNA (Supplementary Table 1), an activity characteristic for diverse antitoxins[21], and cofolding prediction with AlphaFold-Multimer generated high-confidence models (93 average pLDDT, 0.902 iPTM) that support the interaction between the two proteins as a dimer of dimers (Fig. 4b), as commonly observed for type II TAs. Therefore, we hypothesized that DUF6516 is a new toxic TA effector that is neutralized either in *trans* by diverse unrelated antitoxins (subclusters 1–4, 6 and 7) or in *cis* by a fused unknown antitoxin domain (subcluster 5).

To validate the putative TAs experimentally and gain insights into the mechanism of DUF6516-mediated toxicity, we used our established toolbox for TA studies[22]. We targeted TA from six gammaproteobacterial species for testing in *Escherichia coli* surrogate host, and all the putative toxins markedly abrogated *E. coli* growth (Fig. 4d) while the putative antitoxins had no effect (Extended Data Fig. 3). Neutralization assays showed full suppression of toxicity when the toxins were co-expressed with cognate antitoxins (Fig. 4d), thus directly validating that these gene pairs are, indeed, bona fide TA systems.

To probe the mechanism of DUF6516-mediated toxicity, we carried out metabolic labelling assays with $^{35}$S methionine (a proxy for translation), or $^{3}$H uridine (a proxy for transcription) or $^{3}$H thymidine (a proxy for replication). Expression of *Allochromatium tepidum* strain NZ DUF6516 toxin resulted in a decrease in efficiency of $^{35}$S methionine incorporation (Fig. 4e), indicative of the inhibition of protein synthesis. We propose that the effect could be mediated by the yet-unproven RNase activity of the DUF6516 toxin.

We conclude that DUF6516 is a bona fide translation-targeting toxic effector of a new TA family, and propose renaming it TumE (meaning 'dark' in Estonian), with the antitoxin components dubbed as TumA, with A for antitoxin. This example illustrates the difficulty of automating functional annotation for proteins from completely new superfamilies. Here, the combination of genomic context information, remote homology searches on genomic neighbours and deep learning-based structure-guided function prediction helped to formulate a testable functional hypothesis.

## Semantic consistency across the network

Recently, the ProtNLM[10] large language model was implemented as an approach to automatically name proteins in UniProtKB that were titled as uncharacterized proteins. Given that language models have the tendency to 'hallucinate' predictions when faced with an unknown[23], we propose that such an approach would generate a wide diversity of predicted names for completely new protein families. To investigate this hypothesis, we compared the diversity of names predicted by the first release of ProtNLM for proteins in fully dark components and those in fully bright.

In both cases, the distributions of names and words (collectively referred to as semantic diversity) were highly skewed towards extremely low diversities, but the fully dark set was significantly different from the fully bright (Kolmogorov–Smirnov two-sided test statistic 0.2915, $P$ value = $8.882 \times 10^{-16}$, Extended Data Fig. 4a,b). Most bright components had a low semantic diversity, indicating a coherent and consistent naming. The maximum word diversity in these was 37%, corresponding to cases with variations of the same name (for example, several Cytotoxins with different labels for component 100,340). On the other hand, fully dark components tended to have a higher semantic diversity, with a name diversity of 19% (compared to 10% in fully bright) and a word diversity of 7% (compared to 4%). The more consistently named dark components were those with previously submitted names, such as DUF6516.

The dark component with the highest semantic diversity (45%) was component 3,314, composed of 53 proteins with a wide variety of unrelated predicted names, including Integrase, NADH-quinone oxidoreductase subunit F, dynein light chain, prophage protein and so on. Despite this, proteins in component 3,314 share a common fold (Extended Data Fig. 5a) but Foldseek found no hits in the PDB. HHPred searches highlighted a small local match to the tubulin-binding domain of *Chlamydomonas reinhardtii* TRAF3-interacting protein 1 (probability 71%), but when clustered together at sequence level these two groups of proteins only formed a few weak connections (Extended Data Fig. 5a). Although small, component 3,314 is dispersed throughout bacteria and bacteriophages, and the members do not share a conserved genomic context (Extended Data Fig. 5b). Together with the presence of prophage-associated protein encoding genes in these genomic contexts, such as host-nuclease inhibitor protein Gam[24], these data support the prophage protein title.

Another example with a high semantic diversity (35%), and in which structure information aided function assignment, is component 6,732. It consists of 54 entries, some of which are annotated

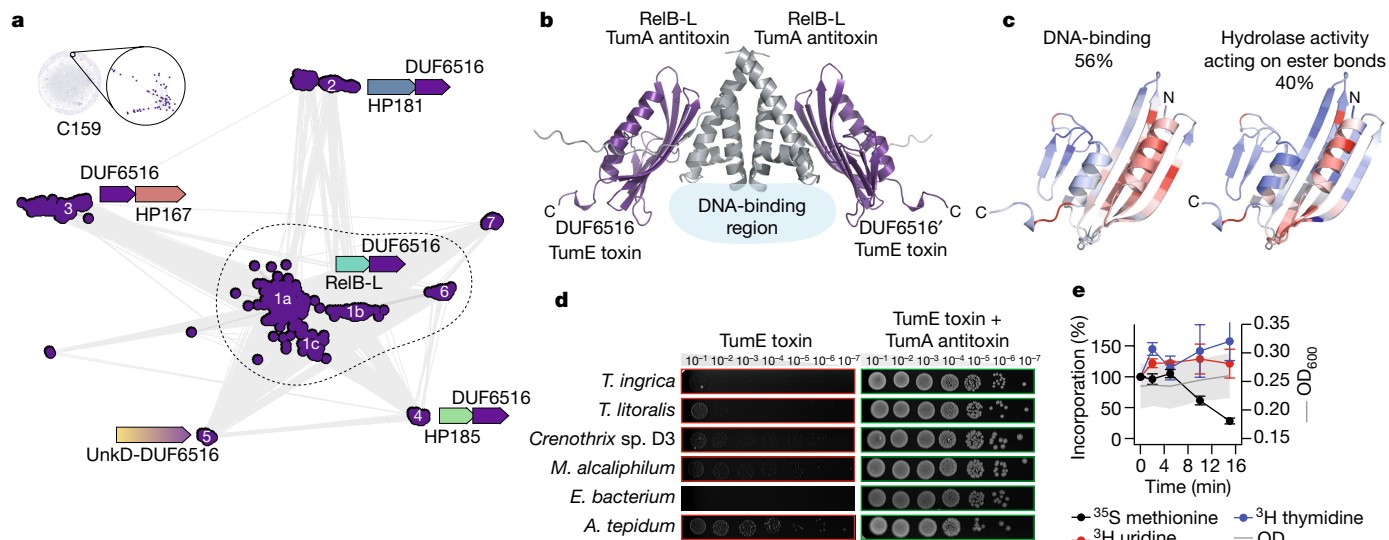

**Fig. 4 | Connected component 159 is a new toxin in the hitherto undescribed TA superfamily TumE–TumA. a**, High-resolution sequence similarity network for 2,453 homologues of the sequences in component 159, computed with CLANS ($E$ value cut-off of $1 \times 10^{-10}$). Points represent proteins and grey lines BLASTp matches ($E$ value $< 1 \times 10^{-4}$). Individual subclusters are labelled 1–7 and subclusters a–c. The consensus genomic contexts, as identified by GCsnap, are shown with different flanking families coloured from blue to red. **b**, A 3D model of the complex between the putative toxin and antitoxin from *A. tepidum* strain NZ, modelled with AlphaFold-Multimer, highlighting the regions where DNA is predicted to interact with the antitoxin. **c**, Structural model of *A. tepidum* TumE/DUF6516 toxin (EntrezID WP_213381069.1) coloured according to the two most frequent molecular functions predicted for 100 homologues with DeepFRI. Residues responsible for the predictions are highlighted in red.

The percentage reflects the frequency of the highlighted prediction. **d**, Validation of *tumE–tumA*. Plasmids for expression of putative toxins (pBAD33 derivates) were cotransformed into *E. coli* BW25113 cells with antitoxin expression plasmids or the empty pMG25 vector. Bacteria were grown for 5 h in liquid LB medium supplemented with appropriate antibiotics and 0.2% glucose. The cultures were normalized to $OD_{600} = 1.0$, serially diluted and spotted on LB plates containing appropriate antibiotics and 0.2% arabinose for toxin induction and 500 μM IPTG for antitoxin induction. The plates were scored after an overnight incubation at 37 °C. **e**, Metabolic labelling assays with *E. coli* BW25113 expressing *A. tepidum* TumE/DUF6516 toxin. Error bars indicate the standard error of the arithmetic mean. All experiments shown in **d** and **e** were performed as $n = 3$ biologically independent replicates (individual independent cultures). All repetitions of the experiments shown in **d** yielded similar results.

inconsistently as AbiEi_1 domain-containing protein, Transposase, Acyl-CoA dehydrogenase and TetR family transcriptional regulator. HHPred searches found no hits in the PDB, but structure-based searches using AFDB models yielded matches to several type II restriction endonucleases. The most similar was EndoMS, a mismatch restriction endonuclease[25] that superposes with a root-mean squared error (r.m.s.d.) of 2.3–2.6 Å. Within the structural alignment, the most conserved residues are those constituting the EndoMS active site (Extended Data Fig. 5c), which are invariant in all members of component 6,732. This suggests that they share a similar active site architecture that has a common restriction endonuclease active site motif (E/D)-Xn-(E/D)XK[26,27], and that component 6,732 may represent a new family of putative restriction endonucleases whose precise function is unknown.

These results highlight that ProtNLM when presented with families with no homologues was indeed hallucinating a diverse range of names. By setting a word diversity cut-off of more than 20% for components with more than 50 proteins, we identified 290 such functionally dark components, covering 4,618 UniRef50 clusters and 37,211 unique protein sequences, and are defining Pfam[2] families for each of them (133 new families available in the next Pfam releases 36.0 and 37.0; Supplementary Table 2). This includes component 3,314 as the PF21779 family and whose members are now titled DUF6874, and component 6,732, which is now PF22187 and its members named DUF6946.

Overall, pooling predictions across the network can help to assess the consistency of automated annotation methods, especially in data-driven approaches. As we define new Pfam families, their naming should become consistent as future versions of ProtNLM consume this data. Starting from UniProt release 2023_01, the criteria for showing ProtNLM names has changed to include an ensemble approach, an increased confidence threshold and an automatic corroboration pipeline

(https://www.uniprot.org/help/ProtNLM), thus many of these hallucinated names have now reverted to being called uncharacterized proteins.

## Structural outliers across the network

Just as semantic diversity revealed novelties in protein sequence space, we also investigated how different the predicted structural characteristics of proteins in our network are from the structures in the PDB. For this, we introduced the concept of structural outliers by using an alphabet of substructure representations covering 1,024 local structural contexts (16 residues in sequence and 10 Å spatial neighbourhood, Extended Data Fig. 6). We trained an outlier detector on PDB structures and predicted that 699,084 AFDB90 structures have substructure compositions that are rare or absent in the PDB, giving us a measure of plausibility that can help to prioritize protein family classification.

Whereas the examples described in the previous section are all structural inliers, we found that 30% of outliers are in dark UniRef50 clusters (Fig. 5a) and that they tend to be shorter and more repetitive than inliers (Fig. 5a,b). Proteins may be structural outliers for a variety of reasons, including new folds as in the next section. Short outliers typically represent fragments of existing families (Fig. 5c), probably due to frameshift errors introduced during whole-genome sequencing. Long outliers tend to be highly repetitive proteins (6,791 clusters, with more than 500 residues and shape-mer diversity fraction less than 0.1, of which 4,948 are bright), which are rare or absent in the PDB (Fig. 5d). Proteins that require conditions to fold that are not modelled by Alpha-Fold2, such as binding partners (Fig. 5e), sometimes have models in AFDB that do not resemble the single chain of the complex as found in the PDB, that is, the predicted monomeric fold may not always be functionally meaningful.

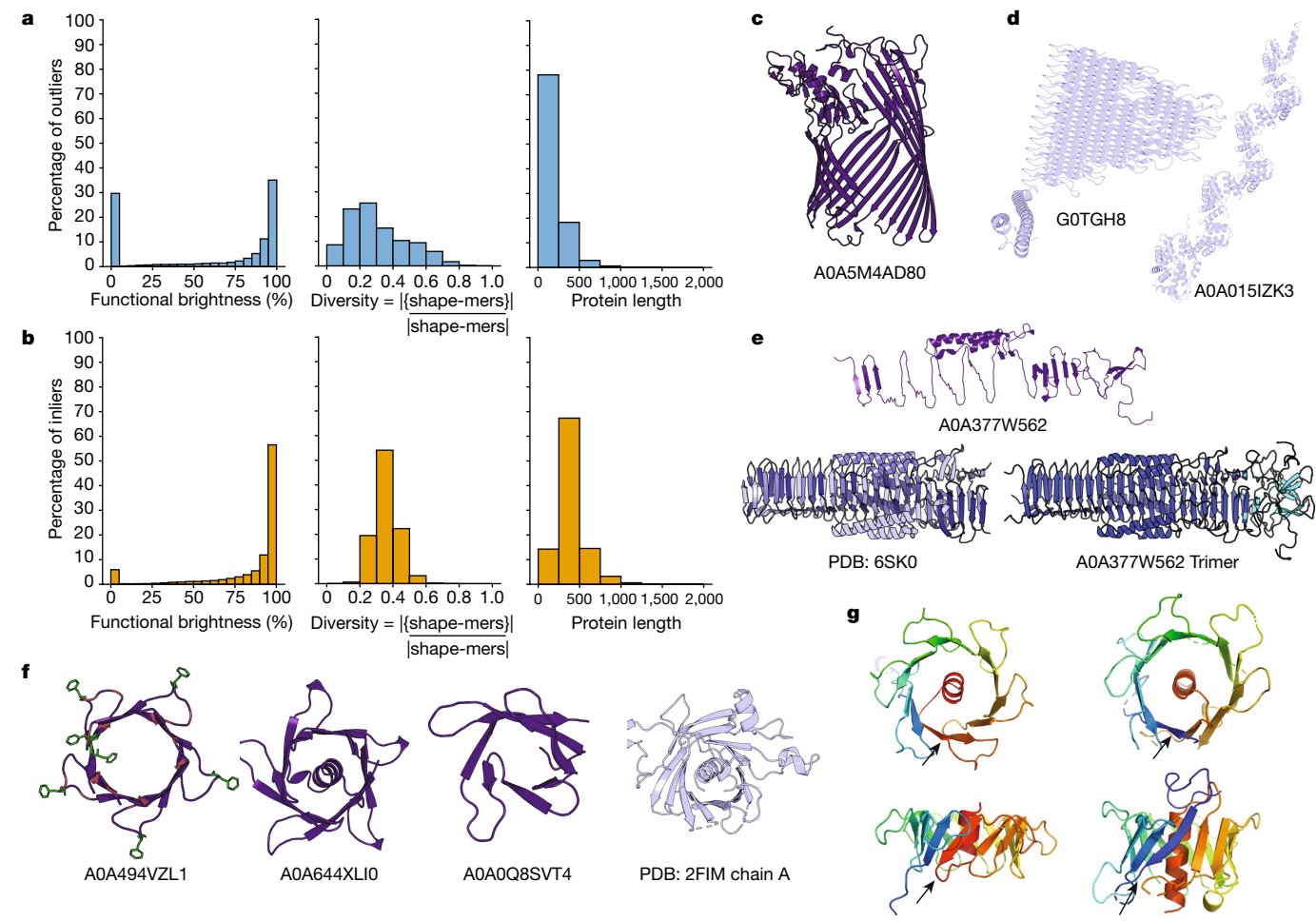

**Fig. 5 | Structural outliers can represent fragments, repetitive proteins, proteins requiring folding conditions out of the scope of AlphaFold2 or new folds. a,b**, Distribution of brightness, shape-mer diversity and length of the structural outliers (**a**) and the same number of structural inliers (**b**) with the most positive outlier scores. Shape-mer diversity is defined as the number of unique shape-mers by the length of the protein. **c**, An AFDB model of TonB-dependent receptor-like protein that is a fragment of the β-barrel domain. More than 16,500 proteins across 1,258 components have this annotation, of which 86% are fully bright. From these, 82% have fewer than the required number of β-sheet shape-mers, despite 55% not being explicitly annotated as fragments in UniProtKB. **d**, Two long repetitive outliers, one belonging to the PE-PGRS superfamily (G0TGH8), thought to be new folds and found widely in mycobacteria[40], and one to the Tetratricopeptide-like helical domain superfamily (A0A015IZK3) in which the median PDB structure length of structures with

resolution less than 3 Å is only 370. **e**, AFDB model annotated as containing 'putative type VI secretion system, Rhs element associated Vgr domain' (A0A377W562), a trimeric PDB structure (PDB ID 6SK0) also containing this domain and an AlphaFold-Multimer model of the A0A377W562 trimer that has 1.1 Å r.m.s.d. to the PDB structure. The AFDB model does not resemble the PDB structure because these proteins form obligate complexes and adopt a trimeric β-solenoid fold. **f**, AlphaFold models of different variations of the β-flower, with positively charged residues in red and phenylalanine in green for A0A494VZL1, and PDB structure of the human Tubby C-terminal domain (PDB ID 2FIM). Black arrows indicate the circularly permuted loop in A0A0S7BXY3 and PDB ID 1ZXU. **g**, AlphaFold model of A0A0S7BXY3 and PDB structure of *Arabidopsis thaliana* putative phospholipid scramblase (PDB ID 1ZXU). Black arrows indicate the circularly permuted loop.

Whereas most fully dark and fully bright components do not contain structural outliers, the outlier content is significantly different between the two sets (Kolmogorov–Smirnov two-sided test statistic 0.0586, $P = 5.245 \times 10^{-81}$, Extended Data Fig. 4c). Fully dark components have on average a higher outlier content (21%) than fully bright (15%), but these only correspond to about half of the structural outliers. Indeed, 44% of outliers are singletons, that is, UniRef50 clusters that do not form a component with at least two nodes, giving us a measure to prioritize even these cases for further analysis, as in the example below.

### The β-flower fold

UniRef50 A0A494VZL1 is an example of a structural outlier that is a singleton in the network. It folds as a shallow, symmetric β-barrel with 96 residues, made of ten short antiparallel β-strands that form a hydrophobic channel. On one side of the β-barrel, the loops connecting each

strand are much longer (nine residues) than those on the other side (four residues). Some are enriched with positively charged arginine and lysine residues, and phenylalanines at the tips pointing towards the exterior of the β-barrel (Fig. 5f). Overall, it looks like a flower (Fig. 5g) and hence we named it the β-flower fold.

Foldseek searches found hits to 43 AFDB90 clusters (TM score of <0.6, most from bacteria) across 13 different components, some of which are bright because they are annotated as Cell wall-binding protein or MORN repeat variant. There are at least three globally different folds (Fig. 5f), differing in the number of strands (8, 10 or 12), with their 'petals' comprising β-hairpins that are arranged in four-, five- or sixfold symmetry. Some of the hits resemble half of a flower, perhaps corresponding to fragments of longer domains, and many enclose a C-terminal hydrophobic α-helix. Some β-flowers also contain N-terminal lipoprotein attachment motifs[28,29], suggesting they may be

associated with the bacterial inner membrane or transferred to the inner leaflet of the outer membrane.

Although no similarity to the PDB was highlighted by Foldseek or HHPred searches, the β-flower folds with sixfold symmetry are reminiscent of the Tubby C-terminal domain[30], which adopts a 12-stranded β-barrel fold enclosing a hydrophobic α-helix (Fig. 5f,g). Tubby-like proteins either bind to phosphoinositides or function as phospholipid scramblases[30]. β-Flowers and Tubby-like proteins share a network of aromatic hydrophobic residues that flank the edges of the β-strands and point towards the interior of the β-barrel, thus engaging in tight contacts with the central hydrophobic helix. The N-terminal strand of Tubby is circularly permuted in β-flowers (Fig. 5g), which leads to a different entry point of the α-helix into the β-barrel channel, and to a difference in its directionality. Furthermore, the length of the β-strands and the connecting loops in the β-flower proteins are notably shorter.

On the basis of their global structural similarity and the presence of a semiconserved [DNEQ]XXG sequence motif at the tip of the β-hairpin, and the repeat unit of both β-flowers and Tubby-like, the diversity of these proteins has been added to Pfam as the new entries PF21784, PF21785 and PF21786, which together with the Tubby C-terminal domain now form the CL0395 clan. This, together with the different types of structural outliers described, highlights that the 3D context provided by the models in AFDB is highly informative for protein analysis efforts and that the structural space covered needs to be put into a coherent evolutionary, functional and local structural context before any model, even with high predicted accuracy, is used as a reference.

## Towards large-scale function annotation

In this work, we carried out a large-scale analysis of the UniProt protein sequence space covered by high-confidence predicted structural models, as made available through AFDB v.4. To aid functional annotation of this space, we constructed an interactive sequence similarity network accounting for about 53 million proteins enriched with predicted name diversity and structural plausibility scores, on a large scale. We demonstrate that this network is a rich source of putative new protein folds, families and superfamilies, providing several starting points for further downstream studies.

We find that many functionally unannotated proteins are remote homologues of annotated ones, relationships that can now be easily explored. Further, more than 1 million proteins belong to completely unannotated connected components, many of which cannot be named consistently using the most recent deep learning-based approaches or contain proteins with structural features distinct from what is seen in the PDB. When combined with traditional protein evolution approaches, structure-based comparisons, genomic context information, structure-based function prediction, and the conservation of local features such as active sites, we could gather support for common evolutionary origins, gain valuable insights into putative functions and put forward concrete testable hypotheses for experimental characterization.

Indeed, the functional annotation of dark proteins, even from a purely computational perspective, requires a combination of data sources and approaches. It is crucial to combine individual predictions across connections in the network to increase the confidence of any hypothesis. Most of our examples had such support from both sequence and structure, and even for the new β-flower fold, a singleton in our network, the presence of a semiconserved sequence motif captured only because of local structural similarities allowed us to generate an initial classification. This information can now help to guide further validation experiments, such as those carried out for TumE.

Our study has some caveats and limitations, however. All alignments required coverage across the entire protein sequence, whereas a domain-based exploration would provide a possible complementary solution. Our functional brightness definition excluded predicted intrinsically disordered and coiled-coil proteins, and misclassifies some functionally uncharacterized proteins as bright due to ambiguous annotations (for example, transmembrane or repeat), or characterized ones as dark due to Putative annotations. Furthermore, we focus only on proteins with high-confidence predicted structures from AFDB, setting aside the wealth of potential darkness in metagenomic data for which structural models are also now available through the ESM (evolutionary scale modelling) Metagenomic Atlas[31]. Although we could already highlight a marked proportion of novelty, in-depth exploration combining many sources of evidence could only be carried out for a few families and folds. Thus, the examples we discuss are the low-hanging fruit of uncharacterized or unannotated protein families, and they are only the tip of the iceberg.

Similarity networks are a common representation of protein space[32,33] and recent approaches to categorize protein diversity and uncover novelties have showcased the importance of incorporating several perspectives and methods in protein annotation[31,34–36]. Our work combines these concepts by providing an annotated similarity network model of protein sequence space on a large scale, which we make available as an interactive and accessible web resource. We anticipate that further advances in deep learning-based methods for function prediction[9], remote homology detection[37,38] and protein structure prediction[31] will allow for analyses on an even larger scale, incorporating more diverse data sources with greater confidence. As such advances continue, we as a community are closer than ever to harnessing the full potential of the protein universe, from unknown biology to new biomedical, pharmaceutical and biotechnological applications.

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

## Methods

### Data collection

We started from the 53,625,855 UniRef50 (ref. 11) clusters as of August 2022 (UniRef v.2022_03) and the 214,683,829 structural models for most UniProtKB entries available through the AFDB (v.4). For each Swiss-Prot[5], TrEMBL[3] and UniParc[12] entry in each UniRef50 cluster we collected their sequence, taxonomy and functional and structural annotations from UniProt and InterPro[6] using custom Python v.3.6 code. Redundant, overlapping annotations were continuously merged (Fig. 1a), selecting as the preferential name the first occurrence that did not include putative, hypothetical, uncharacterized and DUF. Each entry in AFDBv4 was mapped to their UniRef50 cluster, selecting as the structural representative the longest protein with an average pLDDT[41] more than 70.

### Darkness estimation

We define functional brightness of a given protein as the full-length coverage with annotations of its close homologues, with 0% meaning dark and 100% meaning bright. We first computed the full-length coverage with annotations for all entries in all UniRef50 clusters, and considered a cluster as bright as the brightest sequence it encompasses (Fig. 1a). Annotations considered were: domains annotated in InterPro, and families, predicted disorder and predicted coiled-coil regions annotated in UniProtKB and UniParc. All those with putative, hypothetical, uncharacterized and DUF in their name were given a coverage of zero. Pearson correlation was computed using SciPy (v.1.5.4).

### Large-scale sequence similarity network

To model the sequence landscape covered by all UniRef50 clusters with a high-confidence structural model, we built a large-scale sequence similarity network of 6,136,321 clusters having a structural representative with pLDDT more than 90 (AFDB90 dataset). All-against-all MMseqs2 (ref. 13) (release 13-45111) comparisons were carried out with the UniRef50 cluster representatives of all selected clusters, connecting two sequences if they have a match that covers at least 50% of their full-length sequences with an $E$ value better than $10^{-4}$. Each edge was given a weight proportional to the $E$ value of the match, and a maximum of four outbound edges were considered per node (Fig. 1b). The direction of the edges was not further considered.

To visualize the graph, each connected component was simplified to a set of connected communities, detected using the asynchronous label propagation algorithm, as implemented in the asyn_lpa_communities method in networkx (v.2.5.1)[42]. This reduced the graph to a total of 688,852 communities (hereafter referred to as the AFDB90Communities set) connected by 1,488,764 edges, whose layout could then be computed with Cosmograph (https://cosmograph.app/) with the following settings: maximum space allowed 8,192, gravity 0.5, repulsion 1.4, repulsion theta of 1.71, link strength 2, minimum link distance 1 and friction 1. For each community, we collected the longest and median-length representatives, whose structures were used in our analyses. Individual connected components were visualized in figures with Datashader (v.0.12.1, https://datashader.org/index.html).

The interactive, annotated and searchable web version of this network was created using the Cosmograph library (https://github.com/cosmograph-org/cosmos, v.1.3.0) for network visualization and the Mol* toolkit (v.3.35.0)[43] for 3D macromolecular visualization of individual structure representatives. Sequence searches across the interactive network are carried out with a simple $k$-mer search to rapidly identify close homologues in the AFDB (more than 70% sequence identity) and structure searches with Foldseek (3Di method[16], $E$ value better than $10^{-1}$) through its API over the AFDBv4 database filtered to 50% sequence identity (UniProt50). Returned matches are mapped back to their corresponding communities.

### Sequence-based prioritization of dark connected components and their semantic name diversity

Each node in a connected component was attributed a functional brightness value, and components were sorted by their average brightness and their overall size (that is, number of nodes), so that the top ranking were the largest and darkest. To analyse UniProt name diversity, we extracted names as of UniProt v.2022_04 (December 2022, which includes the initial release of ProtNLM[10] predictions) for all UniRef100 representatives included in clusters of fully dark (average functional brightness less than or equal to 5%) and fully bright (average functional brightness more than or equal to 95%) connected components with at least 50 unique protein sequences. We computed the proportion of unique names (that is, name diversity) as well as the proportion of unique words (that is, word diversity), to account for small variations of the same name. Kolmogorov–Smirnov statistical test (two-sided) was computed using SciPy (v.1.5.4).

### Protein substructure decomposition

To represent and analyse 3D substructure composition, we built on Geometricus (v.0.5.0, Python v.3.9)[44], and use 16 rotation invariant moments[45–47] and one chiral invariant moment[48]. These moments were calculated on α-carbon coordinates for overlapping $k$-mers of sizes 8 and 16, and overlapping spheres of radii 5 and 10 Å; for a total of 68 moments for each central residue in a protein, using ProDy (v.2.2.0). We trained a neural network using PyTorch (v.1.12.0)[49] with these 68 moments as input, two linear hidden layers of size 32, a sigmoid output layer with a size of ten and with contrastive loss to reduce the output distance between equivalent pairs of central residues and increase the distance between non-equivalent pairs in a training set. The output of the network for each residue, ten floating point numbers between zero and one, was discretized into 10 bits on the basis of whether the value was greater than or less than 0.5, resulting in 1,024 shape-mers.

The training set was created from structures from the CATH database (v.4.2.0) having less than 40% sequence identity (CATH40) that could be assigned to a CATH functional family (FunFam[50]) with an $E$ value better than $1 \times 10^{-6}$. From these 8,333 structures, US-align (v.20220924)[51] was used to align and superpose all pairs within each FunFam cluster and three randomly chosen pairs for each protein across clusters. Aligned pairs of residues from two same FunFam proteins with TM score more than 0.8 were considered as positive pairs. Aligned or random pairs of residues from two proteins belonging to different CATH superfamilies, with TM score less than 0.6 were considered as negative pairs. In addition, using all 31,883 CATH40 proteins, we sampled up to 50 pairs of central residues from each protein, in which positive pairs had fewer than two sequences distance and negative pairs had 5–20 sequences distance. In total, this resulted in 6 million residue pairs for training, of which 42% were positive pairs. This dataset could be used for training and/or refining any kind of residue-level contrastive learning task. Training took 30 min on one RTX-3080TI with the ADAM optimizer, a batch size of 1,024 and a learning rate of $10^{-3}$ over five epochs.

Shape-mers were calculated for ProteinNet CASP12 proteins in the 100% sequence identity set[52] with more than 20 amino acids. Extended Data Figure 6 shows an example protein with its six most common shape-mers highlighted. We trained a FastText model[53] on the shape-mer bit representations using Gensim[54] (v.4.2.0, window size of 16, embedding size of 1,024). Extended Data Figure 7a shows the sensitivity of SCOPe family retrieval on the SCOPe40 dataset of 11,211 structures for all-versus-all Smith–Waterman alignment with FastText shape-mer similarities used as the score matrix (runtime of 12 min on ten threads). Shape-mer FastText alignment scores are compared to three structure aligners, Dali[55], Foldseek[16] and TM-align[56]; one sequence aligner, MMseqs2 (ref. 13) and two other structure alphabet-based structural sequence aligners, 3D-BLAST[57] and CLE-SW[58], using the scripts and benchmark data provided in van Kempen et al.[16]. Protein-level

embeddings are obtained by averaging across normalized FastText embeddings using the get_sentence_vector function. Extended Data Figure 7b shows the distributions of cosine distances of these embeddings within the same SCOPe family and across SCOPe folds.

### Structural outlier detection

The benchmarking and comparison results (Extended Data Fig. 7) demonstrate that the learned structural alphabet and FastText similarities still have discriminative power in distinguishing protein families, despite being much less local than approaches such as Foldseek and TM-align that work on individual coordinates of up to two residues. We do not explore further alignment optimization, such as compositional bias correction or penalty optimization to increase sensitivity, as more local structural aligners will still have the advantage of higher resolution alignment. However, for the task at hand, our substructure representations give us a good compromise: a discriminative structural alphabet for representing a protein structure as a structural sequence, and substructure decomposition at the level of whole secondary-structural elements allowing for a broader exploration of substructure composition across the AFDB.

For this, we trained the Isolation Forest outlier detection algorithm[59] as implemented in scikit-learn (v.1.1.1)[60] on the ProteinNet CASP12 FastText sentence embeddings with 1% contamination rate. Shape-mers for all AFDB90 structural representative AlphaFold models were calculated following the approach described in the analysis of AFDBv1 (ref. 35) to split each protein into segments with Gaussian smoothed pLDDT more than 70, after first splitting into domains on the basis of a combination of pLDDT and the predicted aligned error matrix, and concatenating shape-mers across each segment in each domain. A shape-mer diversity fraction was defined for each protein as the number of unique shape-mers divided by the total number of residues for which shape-mers are calculated. The trained outlier detection model was used to predict structural outlier scores for AFDB90 proteins. Proteins with negative scores are labelled as outliers. The Kolmogorov–Smirnov statistical test (two-sided) was computed using SciPy (v.1.5.4).

### Computational investigation of selected examples

For the analysis of all examples, we combined data from the sequence-based network and its functional brightness annotations, as well as from structural searches with Foldseek and the outlier scores. Structural homologues for selected representatives (those with a length close to the median length in the component) in the PDB or the AFDB90Communities set were searched with Foldseek (v.7.04e0ec8) using the TM-align mode[16]. Remote sequence homologues were detected for selected representatives by HHPred searches over the PDB, ECOD and Pfam databases through the MPI Bioinformatics toolkit using default settings[61,62]. AlphaFold-Multimer[63] v.3 was used for protein complex prediction when required, with default settings and relaxation, and the model with the best predicted TM score (pTM) and interface pTM score was selected. PyMol (v.2.5.0) was used to visualize selected examples. Further case-by-case analyses were carried out as below.

**Component 27.** All UniRef100 representatives represented by the nodes of connected component 27 were collected and filtered to a maximum sequence identity of 50% with MMseqs2. The reduced set of sequences was aligned with MUSCLE[64] (v.5.1) and the resulting multiple sequence alignment (MSA) used as input for three independent BLASTp[65] searches over the eukaryotic, archaea and bacterial sequences in nr filtered to 70% sequence identity (nr_euk70, nr_arc70, nr_bac70) through the MPI Bioinformatics toolkit as of January 2023. The same BLAST searches were carried out for Swiss-Prot representatives of the PglB, STT3 and YfhO families (UniProt IDs PGLB_CAMJR, STT3_YEAST and YFHO_BACSU). The full-length sequences matched in all searches were then combined with those representatives of connected component 27 and filtered to a maximum sequence identity of 30% with

MMseqs2. The resulting set of 7,004 sequences was clustered on the basis of BLASTp all-against-all searches with CLANS[66] at $E$ value of $1 \times 10^{-20}$ until equilibrium.

**Component 159.** Ninety-four randomly selected sequences from component 159 were aligned with MUSCLE. The resulting alignment was used for three independent PSI-BLAST[65] searches over the eukaryotic, archaea and bacterial sequences in nr (nr_euk, nr_arc, nr_bac) with eight rounds through the MPI Bioinformatics toolkit as of October 2022 (refs. 61,62). All collected sequences were filtered to a maximum sequence identity of 95% with MMseqs2 and clustered on the basis of BLASTp all-against-all pairwise searches with CLANS until equilibrium at $E$ value of $1 \times 10^{-10}$.

The resulting sequence similarity network was used as input for GCsnap (v.1.0.17)[19] for the analysis of the conservation of the genomic contexts encoding for each of the proteins in the individual clusters. A window of four flanking genes was used, MMseqs2 was used for protein family clustering at an $E$ value better than $1 \times 10^{-4}$ and clusters of similar genomic contexts were detected using the operon_cluster_advanced method, which uses PaCMAP (v.0.7.0)[67] to project genomic contexts in two dimensions on the basis of their family composition and DBSCAN[68] (as implemented in scikit-learn v.1.2.2) to identify clusters of similar genomic contexts. Only families that were found in at least 30% of all genomic contexts were considered. For each cluster in the sequence similarity network and each identified neighbour family, up to 100 structure representatives were selected from AFDBv4 and used as input to DeepFRI (v.1.0.0)[9] with default settings. The top ten most common predictions per cluster and/or context family were retrieved. The highest average scoring and most frequently predicted molecular functions were considered the most likely for each case.

We generated the 3D structure of a tetramer consisting of two chains of the *A. tepidum* TumE toxin (EntrezID WP_213381069.1) and two of its putative, cognate TumA antitoxin (EntrezID WP_213381068.1) using AlphaFold-Multimer.

**Component 3,314.** All non-redundant protein sequences represented by the nodes of connected component 3,314 were collected and filtered as for component 27, but over nr filtered to 90% sequence identity (nr_euk90, nr_arc90, nr_bac90, nr_vir90). The same BLAST searches were carried out for the tubulin-binding domain of *Chlamydomonas reinhardtii* TRAF3-interacting protein 1 (UniProt ID A8JBY2_CHLRE, residues 1–131). The full-length sequences matching component 3,314 homologues and the local sequence matching the TRAF3-interacting protein 1 tubulin-binding domain were then combined with representatives of component 3,314 and filtered to a maximum sequence identity of 90% with MMseqs2. The resulting set of 890 sequences was clustered on the basis of BLASTp all-against-all searches with CLANS at $E$ value of $1 \times 10^{-5}$ until equilibrium. The 141 sequences making subcluster 1 in the resulting network, which included the component 3,314-like proteins, were extracted, filtered to a maximum sequence identity of 50% with MMseqs2 and used as input for GCsnap (v.1.0.17), where a window of four flanking genes was used and MMseqs2 used for protein family clustering at an $E$ value better than $1 \times 10^{-4}$.

**Component 6,732.** We have built the Pfam family PF22187 (named DUF6946) using component 6,732 sequences and iteratively searching for homologues using HMMER (v.3.3)[69]. Selected members of this Pfam family were subjected to HHPred searches (HHblits[70] against UniRef30, three iterations with an $E$ value cut-off for inclusion $1 \times 10^{-3}$ for multiple alignment generation and PDB70 search database). Foldseek and Dali server (DaliLite v.5)[55] were subsequently used for structure similarity searches, using AFDB models as queries. The obtained structural alignments were manually inspected and compared with the Pfam family alignment. PF22187 was assigned to clan CL0236 that includes diverse families of nucleases.

**β-flower fold.** We constructed three new Pfam families to cover the sequence space of β-flower proteins. To do this we selected example proteins with four-, five- and sixfold rotational symmetry and iteratively searched for homologues using HMMER's hmmsearch. In general, we used an inclusion threshold of 27 bits, but manually lowered the threshold to identify more homologues or raised it to exclude false matches as identified by AlphaFold2 models. These three families were added to Pfam with accession numbers: PF21784, PF21785 and PF21786 and Pfam clan CL0395, which includes the Tubby C-terminal domain.

## Experimental validation and characterization of a predicted TA family (component 159)

Six Proteobacteria TumE examples from subcluster 1a in the CLANS sequence similarity network produced for component 159 and their cognate TumA antitoxins were selected for experimental characterization (Supplementary Table 3). The plasmids were constructed using the circular polymerase extension cloning (CPEC)[71] approach with synthetic DNA procured from Integrated DNA Technologies. Open reading frames were synthesized with an added strong Shine–Dalgarno sequence (AGGAGGAATTAA) and flanking sequences overlapping with multicloning sites of pBAD33 (ref. 72) (toxin genes) or pMG25 (ref. 73) (antitoxin genes). The DNA fragments were amplified with Phusion polymerase (Thermo Scientific) using pBAD_SD_TOX_fwd and pBAD_TOX_MCS_rev or pMG25_insert_fwd and pMG25_insert_rev primer pairs. pBAD33 was linearized using primers pBAD_lin_1 and pBAD_lin_2 and pMG25 was linearized using pMG25_lin_from_BlpI and pMG25_lin_from_HindIII. CPEC with Phusion polymerase (Thermo Scientific) was performed to clone the genes into the vector backbone (25 cycles with 5 min 30 s extension). The CPEC reaction mixture was transformed into DH5α *E. coli* cells and colony PCR with HOT FIREPol Blend Master Mix (Solis Biodyne) was used to identify colonies with correctly sized inserts. Plasmids were extracted from the overnight cultures using FavorPrepTM Plasmid Extraction Mini Kit (Favorgen) and sequenced. The cognate antitoxin plasmid or empty pMG25 was cotransformed with the toxin plasmids into BW25113 *E. coli* cells. DNA fragments and DNA oligonucleotides used for plasmid construction are provided in Supplementary Table 3.

Validation of toxicity and metabolic labelling experiments with $^{35}$S methionine, $^3$H uridine and $^3$H thymidine were performed as described earlier by Kurata et al.[22]. In brief, *E. coli* BW25113 strains were transformed with a plasmid pair that allowed for controllable co-expression of putative TumE toxins (pBAD33 derivatives, the toxin is expressed under the control of L-arabinose-inducible $P_{BAD}$ promotor) and TumA antitoxins (pMG25 derivatives[73], isopropyl-β-D-thiogalactoside (IPTG)-inducible expression of the antitoxin is driven by $P_{Tac}$ promotor) and pregrown in liquid Luria-Bertani (LB) medium (Lennox) supplemented with 100 μg ml$^{-1}$ carbenicillin (AppliChem) and 25 μg ml$^{-1}$ chloramphenicol (AppliChem) as well as 0.2% glucose (for repression of toxin expression). Serial tenfold 5 μl dilutions were spotted on LB plates supplemented with antibiotics (carbenicillin and chloramphenicol) as well as either 0.2% glucose (repressive conditions) or 0.2% arabinose and 1 mM IPTG (induction conditions). Plates were scored after an overnight incubation at 37 °C.

For metabolic labelling experiments with TumE toxins, *E. coli* BW25113 strains cotransformed with pBAD33 derivatives (for L-arabinose-inducible expression of toxins) as well as the empty pMG25 vector were first plated out on LB plates supplemented with 100 μg ml$^{-1}$ carbenicillin, 25 μg ml$^{-1}$ chloramphenicol and 0.2% glucose (to suppress the leaky expression of the toxin). Using fresh, individual *E. coli* colonies for inoculation, 2 ml of liquid cultures were prepared in defined Neidhardt MOPS minimal media[74] supplemented with 100 μg ml$^{-1}$ carbenicillin, 25 μg ml$^{-1}$ chloramphenicol, 0.1% of casamino acids and 0.2% glucose, and grown overnight at 37 °C with shaking. Next, experimental 15 ml cultures were prepared in 125 ml conical flasks in MOPS

medium supplemented with 0.5% glycerol, 100 μg ml$^{-1}$ carbenicillin, 25 μg ml$^{-1}$ chloramphenicol as well as a set of 19 amino acids (lacking methionine), each at final concentration of 25 μg ml$^{-1}$. These cultures were inoculated overnight to final optical density (OD$_{600}$) of 0.05, and grown at 37 °C with shaking up to of OD$_{600}$ of 0.2. At this point, one 1 ml aliquot (the pre-induction zero time-point) was transferred to 1.5 ml Eppendorf tubes containing 10 μl of radioisotope—$^{35}$S methionine (4.35 μCi, Perkin Elmer), $^3$H uridine (0.65 μCi, Perkin Elmer) or $^3$H thymidine (2 μCi, Perkin Elmer)—and transferred to the heat block at 37 °C. Immediately afterwards, the expression of toxins in the remaining 14 ml of culture was induced by addition of L-arabinose (final concentration of 0.2%). Throughout the toxin induction time course, 1 ml aliquots were taken from the 15 ml of culture and transferred to 1.5 ml Eppendorf tubes containing 10 μl of radioisotope ($^{35}$S methionine, $^3$H uridine or $^3$H thymidine). The incorporation of radioisotopes was stopped after 8 min of incubation at 37 °C by adding 200 μl of ice-cold 50% trichloroacetic acid to 1 ml of culture. In parallel with taking the time-points for labelling, 1 ml aliquots were taken for OD$_{600}$ measurements. Isotope incorporation was quantified by normalizing radioactivity counts per million to OD$_{600}$, with the pre-induction zero time-point set at 100%.

All experiments were performed in three biological replicates (that is, using three independent cultures inoculated from three different colonies).

## Reporting summary

Further information on research design is available in the Nature Portfolio Reporting Summary linked to this article.

## Data availability

All data used for this study are publicly available in UniProtKB (https://www.uniprot.org/, UniRef v.2022_03), the AFDB (https://alphafold.ebi.ac.uk/, v.4, with specific examples corresponding to UniProt IDs A0A0E3S9F7, A0A3R7AQ40, A0A520JWH3, A0A1W9UY89, A0A7J4P9B0, A0A0F9A5W1, A0A0P9GTS8, A0A418VYX3, A0A2S5M855, A0A2K2VML8, A0A098EYB0, G0TGH8, A0A015IZK3, A0A377W562, A0A494VZL1, A0A0S7BXY3, A0A7X7MB17, YFHO_BACSU, A8JBY2_CHLRE and A0A3A8FAL8), the CATH database (https://www.cathdb.info/, v.4.2.0), ProteinNet (https://github.com/aqlaboratory/proteinnet, CASP12 dataset), Foldseek benchmark data (https://wwwuser.gwdg.de/~compbiol/foldseek), the PDB (https://www.ebi.ac.uk/pdbe/, PDB IDs 5FMT, 5GKH, 8D3P, 6SK0, 2FIM, 1ZXU, 6GXC and 7OCI) and National Center for Biotechnology Information GenBank (https://www.ncbi.nlm.nih.gov/protein/, EntrezIDs WP_213381069.1 and WP_213381068.1). For the laboratory experiments, all data generated are included in the paper and Supplementary materials. All data and metadata generated supporting the large and the individual sequence similarity networks are available at https://zenodo.org/record/8121336 (CC-BY 4.0). An interactive version of the large sequence similarity network, queryable by keyword, UniProt ID, connected component ID, community ID, protein sequence and protein structure, is available at https://uniprot3d.org/atlas/AFDB90v4. The interactive resource allows also for the downloading of the metadata associated with each individual connected component and community, as well as for the results of any search. Source data are provided with this paper.

## Code availability

All the code to collect and process the annotation data in UniProtKB, UniParc and InterPro, and the pLDDT data from AFDB are available at https://github.com/ProteinUniverseAtlas/dbuilder. Model and training code for shape-mer generation can be found in https://github.com/TurtleTools/geometricus/tree/master/training. All analysis code, including that to process the large sequence similarity network,

decompose structures and generate the plots shown, is available at https://github.com/ProteinUniverseAtlas/AFDB90v4 (Apache).

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

**Acknowledgements** We thank the SWISS-MODEL development team for technical support and text revisions, A. Rustamova for helping with metabolic labelling experiments, G. C. Atkinson, T. Kościółek, L. Lane, M. Bileschi and L. Regan for insightful discussions and comments, the Cosmograph team for providing the fastest network graph visualization tool that works in the browser and sciCORE at the University of Basel (https://scicore.unibas.ch/) for providing computational resources and system administration support. This work was supported by funding from the Swiss Institute of Bioinformatics (https://www.sib.swiss/), the Biozentrum of the University of Basel (https://www.biozentrum.unibas.ch/), by the European Union by the project MIBEst H2020-WIDESPREAD-2018-2020/GA number 857518 (T.T. and V.H.), by a grant from the Estonian Research Council (no. PRG335 for T.T. and V.H.), the Knut and Alice Wallenberg Foundation (grant no. 2020-0037 to V.H.), Swedish Research Council (Vetenskapsrådet) grants (no. 2021-01146 to V.H.), Cancerfonden (no. 20 0872 Pj to V.H.) and the Biotechnology and Biological Sciences Research Council and the National Science Foundation Directorate for Biological Sciences (no. BB/X012492/1 to A.B.).

**Author contributions** J.P. and J.D. conceptualized the study. J.P. performed the functional darkness analysis and constructed the sequence-based network. J.D. performed the structure outlier analysis. A.M.W. developed the interactive web resource and J.P., J.D. and G.T. coordinated its development. J.P., J.D., A.B. and A.A. performed the computational analysis of selected examples. G.S., M. Akdel, J.P., J.D. and A.M.W. developed computational methodologies. T.M., T.B. and M. Abdullah carried out wet-laboratory experiments. V.H. and T.T. conceptualized, coordinated and supervised wet-lab experiments. T.S., A.B., V.H., T.T., G.T. and J.P. acquired funding. J.P. and J.D. wrote the original draft. All authors contributed, reviewed, edited and approved the manuscript.

**Competing interests** The authors declare no competing interests.

**Additional information**
**Correspondence and requests for materials** should be addressed to Torsten Schwede or Joana Pereira.

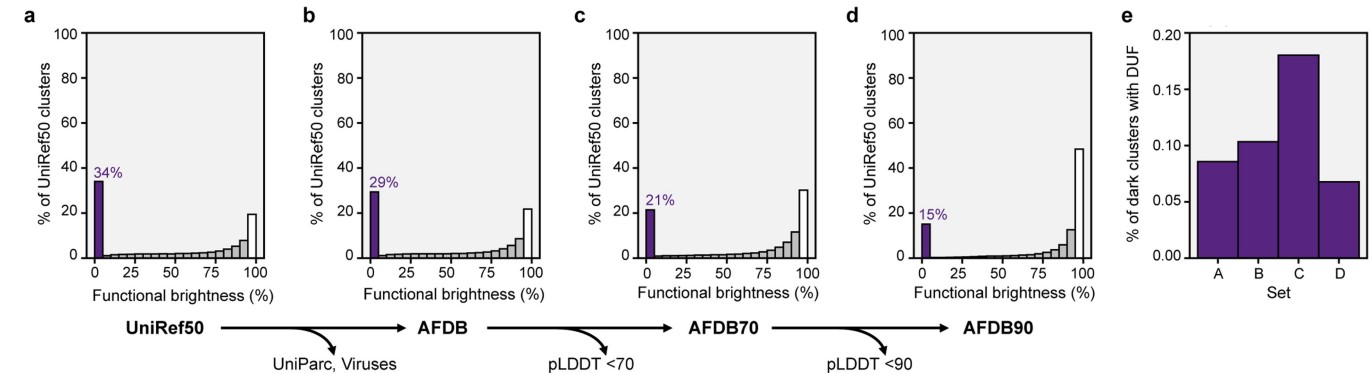

**Extended Data Fig. 1 | Distribution of functional darkness in UniProt and AFDB (version 4).** Functional brightness distribution in (a) UniRef50, (b) UniRef50 clusters with models in AFDB (which excludes long proteins, and those UniRef50 clusters composed solely of UniParc entries and viral proteins), (c) UniRef50 clusters whose best structural representative has an average pLDDT > 70, and (d) UniRef50 clusters whose best structural representative has an average pLDDT > 90. For each set, the percentage of fully dark UniRef50 clusters, and corresponding brightness bin, are highlighted in purple. The bar associated with functionally bright UniRef50 clusters (functional brightness >95%) is marked in white. (e) Percentage of fully dark UniRef50 clusters with proteins annotated as a domain of unknown function (DUF) in each set a-e.

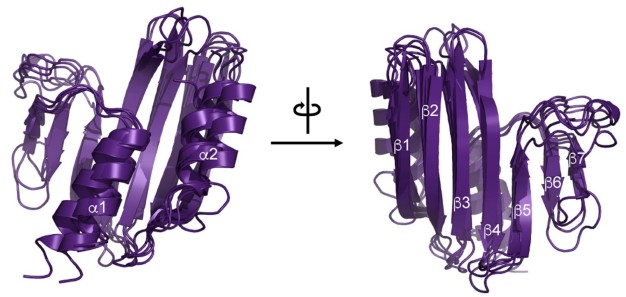

**Extended Data Fig. 2 | Structural conservation and structure-based function prediction of TumE.** Structural superposition of five randomly selected members of component 159 (UniProt IDs A0A0E3S9F7, A0A3R7AQ40, A0A520JWH3, A0A1W9UY89, A0A7J4P9B0) with secondary structure elements labelled.

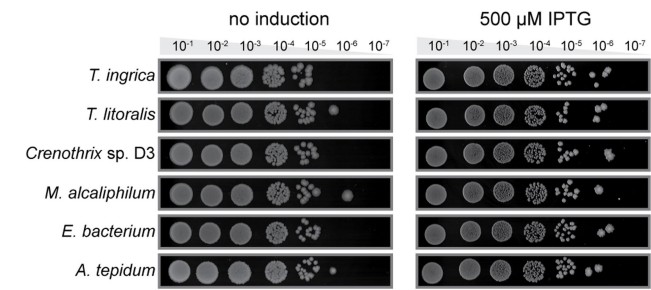

**Extended Data Fig. 3 | Testing the toxicity of putative TumA antitoxins.**
Antitoxin expression plasmids were cotransformed with empty toxin expression vectors (pBAD33) into *E. coli* BW25113 cells. The bacterial cultures were started from a single colony and grown for five hours in liquid LB media supplemented with appropriate antibiotics. The cultures were normalised to $OD_{600} = 1.0$, serially diluted and spotted on LB agar plates containing appropriate antibiotics and 500 μM IPTG for antitoxin induction and 0.2% arabinose to mimic the conditions in toxin neutralization assay. The experiment was made in n = 3 biologically independent replicates.

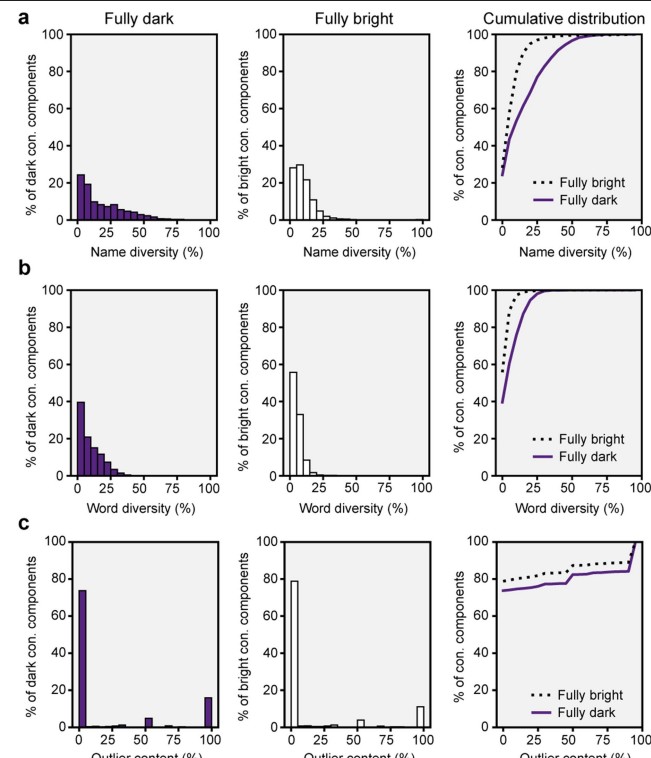

**Extended Data Fig. 4 | Diversity of the (a) names predicted by ProtNLM and (b) their word composition, as well as the (c) fraction of structural outliers, for all fully dark and fully bright connected components.** Name diversity is calculated as the number of unique protein names within a component by the total number of component proteins. Word diversity is calculated as the number of unique words across all protein names within a component by the total number of words, ignoring the words "protein", "domain", "family", "containing", and "superfamily". Outlier content is calculated as the percentage of UniRef50 clusters with negative structural outlier scores within that component. Fully bright and fully dark distributions were compared using a two-sided Kolmogorov–Smirnov test, resulting in a test statistic of 0.2915 and P-value = $8.8829 \times 10^{-16}$ for (b) and test statistic 0.05859 and P-value = $5.245 \times 10^{-81}$ for (c).

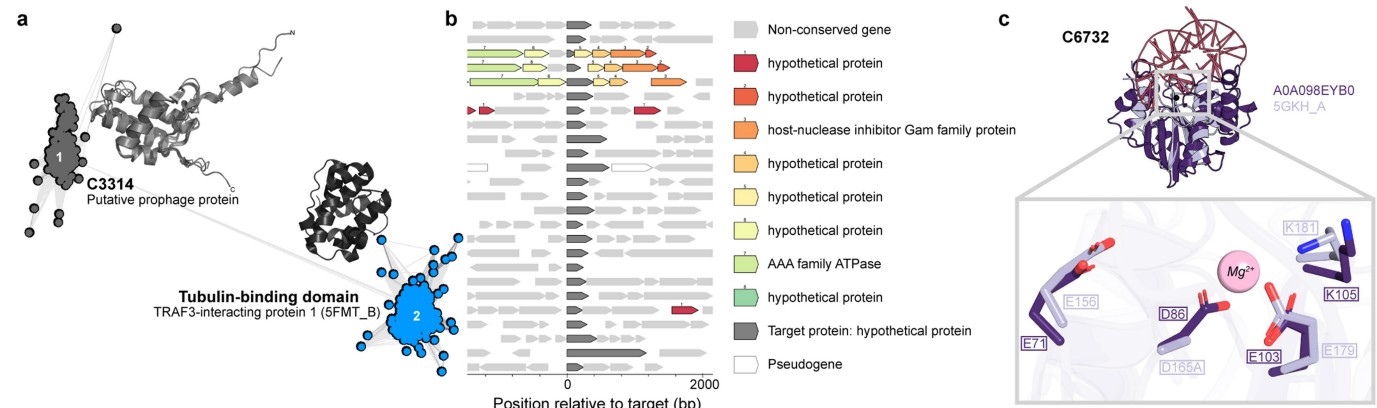

**Extended Data Fig. 5 | The highly semantically diverse prophage-associated connected components 3,314 and 6,732.** (a) Sequence similarity network of homologs of members of connected component 3,314 and the tubulin-binding domain of TRAF3-interacting protein 1, as computed with CLANS at an E value threshold of $1 \times 10^{-5}$. Points represent individual proteins and grey lines BLASTp matches at an E-value better than $1 \times 10^{-4}$. Individual subclusters are labelled 1-2 and structural representatives are shown. For subcluster 1, 5 randomly selected structural representatives of component 3,314 are superposed (UniProt IDs A0A0F9A5W1, A0A0P9GTS8, A0A418VYX3, A0A2S5M855, A0A2K2VML8). For subcluster 2, the tubulin-binding domain of *Chlamydomonas reinhardtii*

TRAF3-interacting protein 1 (PDB ID 5FMT, chain B) is shown. (b) Genomic context conservation of 30 sequences from subcluster 1 with a maximum sequence identity of 30%, as computed with GCsnap. (c) Structure superposition of component 6,732 representative (A0A098EYB0, purple) and mismatch restriction endonuclease EndoMS (PDB ID 5GKH, chain A, grey). The grey box indicates the active site pocket with conserved residues labelled. Note that the residue D165 corresponding to D86 is mutated to alanine in the PDB structure. Structural homologs were searched both with Foldseek, which resulted in a hit to Cas4 endonuclease PDB ID 8D3P with TM-score 0.34, and Dali[55] multiple hits to restriction endonucleases, the top-ranking with a Z-score of 8.2.

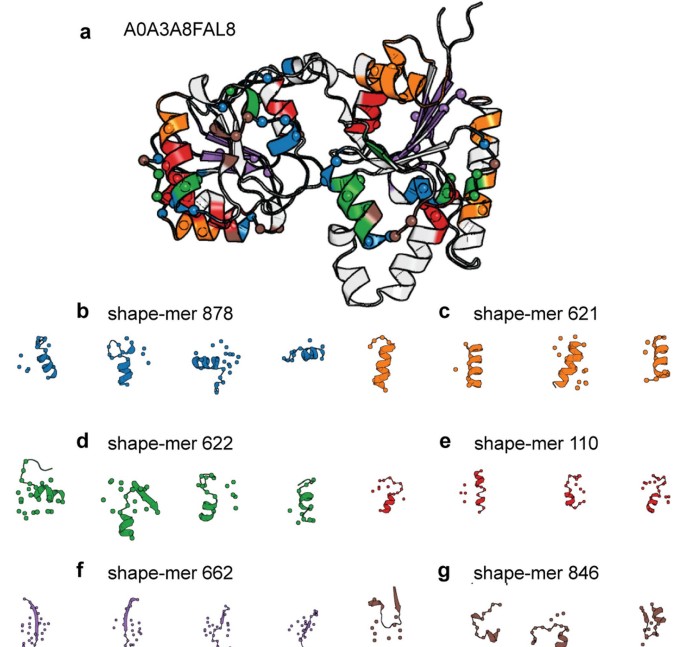

**a** A0A3A8FAL8

**b** shape-mer 878  **c** shape-mer 621

**d** shape-mer 622  **e** shape-mer 110

**f** shape-mer 662  **g** shape-mer 846

**Extended Data Fig. 6 | An example of substructure decomposition.**
(a) An example AlphaFold protein model with its 6 most common shape-mers highlighted in different colours. Spheres mark the shape-mer central residue and backbone atoms within 4 Å are coloured. (b-g) Four random representatives of each selected shape-mer, obtained from CATH proteins with <20% sequence identity. Spheres depict positions within 8 residues in sequence and 10 Å spatially from the central residue.

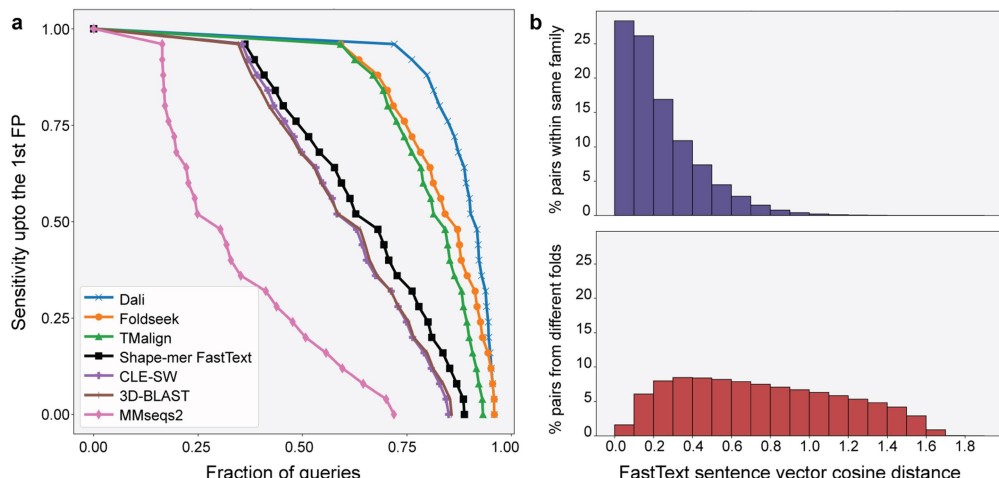

**Extended Data Fig. 7 | Shape-mer representations combined with FastText can discriminate between protein families.** (a) Cumulative distributions of sensitivity for homology detection on the SCOPe40 database of single-domain structures. True positives (TPs) are matches within the same SCOPe family, false positives (FPs) are matches between different folds. Sensitivity is the area under the ROC curve up to the first FP. Results based on shape-mer FastText Smith-Waterman alignment are shown in black. (b) Protein-level embedding distance measured as the cosine distance of FastText sentence vectors for proteins within the same SCOPe family (top) and from different SCOPe folds (bottom).

Torsten Schwede

# Reporting Summary

## Statistics

For all statistical analyses, confirm that the following items are present in the figure legend, table legend, main text, or Methods section.

| n/a | Confirmed | |
|---|---|---|
| ☐ | ☒ | The exact sample size ($n$) for each experimental group/condition, given as a discrete number and unit of measurement |
| ☐ | ☒ | A statement on whether measurements were taken from distinct samples or whether the same sample was measured repeatedly |
| ☐ | ☒ | The statistical test(s) used AND whether they are one- or two-sided<br>*Only common tests should be described solely by name; describe more complex techniques in the Methods section.* |
| ☒ | ☐ | A description of all covariates tested |
| ☒ | ☐ | A description of any assumptions or corrections, such as tests of normality and adjustment for multiple comparisons |
| ☐ | ☒ | A full description of the statistical parameters including central tendency (e.g. means) or other basic estimates (e.g. regression coefficient) AND variation (e.g. standard deviation) or associated estimates of uncertainty (e.g. confidence intervals) |
| ☐ | ☒ | For null hypothesis testing, the test statistic (e.g. $F$, $t$, $r$) with confidence intervals, effect sizes, degrees of freedom and $P$ value noted<br>*Give P values as exact values whenever suitable.* |
| ☒ | ☐ | For Bayesian analysis, information on the choice of priors and Markov chain Monte Carlo settings |
| ☒ | ☐ | For hierarchical and complex designs, identification of the appropriate level for tests and full reporting of outcomes |
| ☐ | ☒ | Estimates of effect sizes (e.g. Cohen's $d$, Pearson's $r$), indicating how they were calculated |

*Our web collection on statistics for biologists contains articles on many of the points above.*

## Software and code

Policy information about availability of computer code

| | |
|---|---|
| Data collection | Data collection for the annotated network was carried out using custom code available at https://github.com/ProteinUniverseAtlas/dbuilder which uses Python 3.6 and PyMongo v3.11.3.<br>Training data for protein substructure decomposition and outlier detection was created using custom code available at https://github.com/TurtleTools/geometricus/tree/master/training using Python 3.9, cath-tools-genomescan (version 17/12/2019), and ProteinNet (CASP12 dataset) |
| Data analysis | Custom code for data analysis can be found at https://github.com/ProteinUniverseAtlas/AFDB90v4 and uses:<br>Python 3.6, 3.9<br>SciPy (v1.5.4)<br>NetworkX (v2.5.1)<br>ProDy (v2.2.0)<br>Geometricus (v0.5.0)<br>PyTorch (v1.12.0)<br>Gensim (v4.2.0)<br>scikit-learn (v1.1.1)<br>Datashader (v0.12.1)<br><br>In addition, the following tools were used for analyses as described in the Methods:<br>MMseqs (release 13-45111)<br>MUSCLE (v5.1)<br>GCsnap (v1.0.17) |

DeepFRI (v1.0.0)
Foldseek (Version: 7.04e0ec8)
AlphaFold (v2.3.0)
PyMol (open-source v2.5.0)

For manuscripts utilizing custom algorithms or software that are central to the research but not yet described in published literature, software must be made available to editors and reviewers. We strongly encourage code deposition in a community repository (e.g. GitHub). See the Nature Portfolio guidelines for submitting code & software for further information.

## Data

Policy information about availability of data

All manuscripts must include a data availability statement. This statement should provide the following information, where applicable:
- Accession codes, unique identifiers, or web links for publicly available datasets
- A description of any restrictions on data availability
- For clinical datasets or third party data, please ensure that the statement adheres to our policy

All data used for this study is publicly available in UniProtKB (https://www.uniprot.org/, UniRef version 2022_03), the AlphaFold database (https://alphafold.ebi.ac.uk/, version 4, with specific examples corresponding to UniProt IDs A0A0E3S9F7, A0A3R7AQ40, A0A520JWH3, A0A1W9UY89, A0A7J4P9B0, A0A0F9A5W1, A0A0P9GTS8, AOA418VYX3, A0A2S5M855, A0A2K2VML8, A0A098EYBO, G0TGH8, A0A015IZK3, A0A377W562, A0A494VZL1, A0A0S7BXY3, A0A7X7MB17, YFHO_BACSU, A8JBY2_CHLRE, and A0A3A8FAL8), the CATH database (https://www.cathdb.info/, version 4.2.0), ProteinNet (https://github.com/aqlaboratory/proteinnet, CASP12 dataset), Foldseek benchmark data (https://wwwuser.gwdg.de/~compbiol/foldseek), the Protein Data Bank (https://www.ebi.ac.uk/pdbe/, PDB IDs 5FMT, 5GKH, 8D3P, 6SK0, 2FIM, 1ZXU, 6GXC and 7OCI), and NCBI GenBank (https://www.ncbi.nlm.nih.gov/protein/, EntrezIDs WP_213381069.1 and WP_213381068.1).
For the laboratory experiments all data generated are included in the manuscript and supplementary materials. All data and metadata generated supporting the large and the individual sequence similarity networks are available at https://zenodo.org/record/8121336 (CC-BY 4.0). An interactive version of the large sequence similarity network, queryable by keyword, UniProt ID, connected component ID, community ID, protein sequence, and protein structure, is available at https://uniprot3d.org/atlas/AFDB90v4. The interactive resource allows also for the downloading of the metadata associated with each individual connected component and community, as well as for the results of any search.

## Research involving human participants, their data, or biological material

Policy information about studies with human participants or human data. See also policy information about sex, gender (identity/presentation), and sexual orientation and race, ethnicity and racism.

| | |
|---|---|
| Reporting on sex and gender | Not applicable. No human participants or human data was used in this study. |
| Reporting on race, ethnicity, or other socially relevant groupings | Not applicable. No human participants or human data was used in this study. |
| Population characteristics | Not applicable. No human participants or human data was used in this study. |
| Recruitment | Not applicable. No human participants or human data was used in this study. |
| Ethics oversight | Not applicable. No human participants or human data was used in this study. |

Note that full information on the approval of the study protocol must also be provided in the manuscript.

# Field-specific reporting

Please select the one below that is the best fit for your research. If you are not sure, read the appropriate sections before making your selection.

☒ Life sciences          ☐ Behavioural & social sciences          ☐ Ecological, evolutionary & environmental sciences

For a reference copy of the document with all sections, see nature.com/documents/nr-reporting-summary-flat.pdf

# Life sciences study design

All studies must disclose on these points even when the disclosure is negative.

| | |
|---|---|
| Sample size | We followed the standard practices in the toxin-antitoxin molecular microbiology field. The sample size of three is regularly used in toxin-antitoxin microbiology field for growth and metabolic labeling experiments. The effects were strong and do not require further statistical analysis. |
| Data exclusions | No data were excluded. |
| Replication | The experiments were repeated in at least three biological independent replicates. All of the attempts were successful and showed the same results. |

| Randomization | We followed the standard practices in the toxin-antitoxin molecular microbiology field. Randomization of samples is generally not practiced. |
| Blinding | We followed the standard practices in the toxin-antitoxin molecular microbiology field. Blinding is generally not practiced. |

# Reporting for specific materials, systems and methods

We require information from authors about some types of materials, experimental systems and methods used in many studies. Here, indicate whether each material, system or method listed is relevant to your study. If you are not sure if a list item applies to your research, read the appropriate section before selecting a response.

## Materials & experimental systems

| n/a | Involved in the study |
|-----|-----------------------|
| ☒ ☐ | Antibodies |
| ☒ ☐ | Eukaryotic cell lines |
| ☒ ☐ | Palaeontology and archaeology |
| ☒ ☐ | Animals and other organisms |
| ☒ ☐ | Clinical data |
| ☒ ☐ | Dual use research of concern |
| ☒ ☐ | Plants |

## Methods

| n/a | Involved in the study |
|-----|-----------------------|
| ☒ ☐ | ChIP-seq |
| ☒ ☐ | Flow cytometry |
| ☒ ☐ | MRI-based neuroimaging |

