## [Peer Review File · Nature]

Manuscript Title: Uncovering new families and folds in the natural protein universe

Reviewer Comments & Author Rebuttals

Reviewer Reports on the Initial Version:

Referee expertise:

Referee #1: computational structure/function analysis, structural bioinformatics, protein family evolution

Referee #2: computational analysis of protein family evolution

Referees' comments:

Referee #1:

A. Summary of the key results

This work presents a large-scale clustering of available protein sequences from UniProt50 (the UniProt database clustered to a maximum of 50% sequence identity) to identify uncharacterized families, i.e. groups of similar sequences that have not yet been functionally annotated. The uncharacterized families were defined as those that generally lack sequence-based annotations (domains, disorder, coiled coils, etc.) provided by the InterPro, UniProtKB, and UniParc databases, and were subsequently analyzed on a case-by-case basis, resulting in their better characterization.

B. Originality and significance

The authors present a solid bioinformatics study supported by experimental validation of some of the results (TumE-TumA toxin-anitotxin system, Fig. 5). While the paper is methodologically sound and takes advantage of all the recent advances in protein bioinformatics, I have some concerns about the study design, the main conclusions, and the availability of the data.

The authors point out (pp. 3-4, lines 119-121) that some of the uncharacterized families have corresponding high-quality AF2 models and some do not. Given this, it is unclear why the similarity network (Fig. 1) was computed only for UniRef50 clusters containing a high-quality model in the AFDB - first, the clustering is purely sequence-based (no structural information is required), second, as mentioned above, there are uncharacterized families for which the structural information is missing in the AFDB (this, however, does not mean that it was impossible to compute a model for them, but that it is missing in the AFDB).

The first example (uncharacterized family/component 27, p. 5) further highlights the fact that the AF2 models were not really necessary for the annotation. An HHpred search with a representative

sequence (A0A7X7MB17) clearly shows the homology to glycotransferases (STT3/PglB/AgIB) with very good e-values. In this case, structural searches were not essential (p. 5, lines 167-174) and could even be misleading, as I will try to explain below. The authors write: "This example illustrates the power of structural alignment to resolve undetected remote homology relationships". The structural search did identify other similar proteins (Fig. 4b), but I would be very cautious about claiming homology based on structural similarity alone. There are examples of structures that are similar, but the extent of their homology is debated (e.g. the class of "immunoglobulin-like"). With this in mind, I believe that Fig. 4a should include only sequence-based homologs. Otherwise, it suggests an equivalence of sequence and structural similarity in homology detection (which may be true in this case, but is not a general rule).

In contrast to the case study mentioned above, the analyses of "Structural outlier diversity in AFDB90" (p. 7 ff.) make use of the AFDB structural data, but do not seem to be related to the sequence-based clustering procedure of the AFDB90 subset. In other words, these studies, including the discovery of the beta-flower fold, could easily have been made into a separate paper, suggesting their rather loose connection to the rest of the work.

The above considerations lead to a main question about the main purpose of the work presented. Is it to provide starting points (i.e., uncharacterized families) for analyses such as those exemplified (pp. 5-9)? If this is the case, then first, as mentioned above, I trust that the analyses should not be limited to proteins that have counterparts in the AFDB, and second, the data should be provided in a more readable form. Currently the only options are a huge 4GB file from Zenodo or the interactive AFDB90v4 atlas (see also point F).

C. Data & methodology

The authors relied on state-of-the-art and well-established approaches to sequence and structure analysis. However, I would expect the data derived from metagenomes to be considered, as they may contain many "dark" families.

D. Appropriate use of statistics and treatment of uncertainties

Not applicable.

E. Conclusions: robustness, validity, reliability

Page 10, lines 401-403, "Our results indicate that 19% of this space is composed of protein families and superfamilies that are functionally dark and cannot, solely on the basis of sequence similarity, be annotated to a known protein family": The procedures for using structural similarity to detect homology are not fully established, and there are clearly cases where such similarity may not be the result of common ancestry. I think this should be addressed and discussed more directly in the paper, especially if the AFDB models remain the focus of the study.

Page 10, lines 406-408, "Our findings highlight the power of large-scale similarity networks to curate protein function. In cases where simple homology transfer methods are not effective, pooling

information from across the network can enhance remote homology detection”: I am not sure which parts of the work the authors are referring to. The "large-scale similarity network" allows the detection of poorly annotated protein families, which can then be analyzed on a case-by-case basis, however I can't see how "pooling information from across the network can enhance remote homology detection". Authors are encouraged to be more specific in this regard.

F. Suggested improvements: experiments, data for possible revision

The atlas is visually appealing, but has limited functionality, and the 4GB flat file can only be used by experienced users. What I miss is an online resource that would allow easy searching (by sequence or structure) and browsing of the identified families (both well and poorly annotated). Such a resource could stimulate more analyses like those presented by the authors. Currently, I see no easy way to access the results of "the discovery of at least 281 putative new protein families" (p. 2, line 66), except those listed in Table S1.

G. References

The bibliography seems adequate. However, the lack of reference to ESMFold and ESM Atlas is very surprising. The ESM Atlas covers much more sequence space than the AFDB, and perhaps there is more "darkness" in it?

H. Clarity and context

The paper, including the abstract, is well written and clear. In general, however, it may be misleading in suggesting direct use of the AF2 models: "we measured the extent to which AlphaFold has illuminated the <dark matter> of the natural protein universe, and modelled the diversity of the proteins it covers". As I discussed above, the AFDB may be helpful/essential in analyzing some of the identified uncharacterized families, but its entire content (>200 million structures) is not really the focus of the work.

In its current form, the paper promises a combined structure-sequence study of protein "dark matter", but in fact provides a sequence-based pipeline for identifying unannotated ("dark") protein families. Such a pipeline is not very novel, given the availability of methods such as MMseqs2 and DIAMOND, and could be considered a simple pre-screening step for each of the case studies presented (regardless of whether it ultimately requires the use of structural predictions, as in the case of beta-flower, or not, as in the case of glycotransferases). This brings up another important point. While each of the case studies is interesting and well presented, they do not form a coherent story that supports the main promises of the paper (using the whole AFDB to uncover protein "dark matter"), leaving the impression that the submitted work is a compilation of a few good but independent projects.

Referee #2 (Remarks to the Author):

We are entering the era of comprehensive coverage of protein sequence-structure-function space

due to revolutionary progress in protein structure prediction. This very timely and exciting manuscript capitalizes on these developments. It offers a comprehensive clustering of ~200 million protein structure models in the AF database and skims through the highlights of the findings.

This work is significant because most researchers do not have the computational capabilities to access the entire AF database and explore it for protein families of their interest. This work equips them with an easier way to access the data. In my opinion, this is the most significant value of this work for the research community.

Furthermore, the paper stands on its own, making a unique contribution to evolutionary protein classification and structure-function relationship. The distribution of protein clusters and functional annotations among them, as well as other general statistics reported in the paper, are illuminating and instructive. Biological discoveries (one is validated experimentally) are interesting. For all these reasons, I believe this paper will be noted by a broad audience and widely cited.

The methods used are solid and are described in detail so that they can be reproduced.

The following improvements would further increase the impact of this work.

1. Access to the data and clusters: Many biologists with developing computational skills would like to access and explore protein clusters obtained in this work. Therefore, it would be helpful to explain more about what is offered for such exploration and how it should be done. I see that all the data are offered for download, but it takes some computational expertise (really!) to figure out what is what. Simplifying this process (e.g., answering questions such as: how to find which cluster my favorite protein belongs to and what functional annotations are associated with this cluster) would help. To be clear, I am not asking for much work here, but for some supplementary file (kind of a simplified and easily accessible “readme”) that explains what can be downloaded and how and answers some possible FAQs about these datasets.

2. Some sections in the results are riddled with numbers and read like free-text tables. Is it necessary? I personally was a bit bored reading these sections. Can they be rephrased somewhat and made more interesting, with these texts transferred to the supplement? I’m not insisting on it, it is just my opinion, and I understand that the authors put much work into getting these numbers, and they must be proud of what they got.

3. A comment which should not detract from the value of this study but seems like an interesting result that (possibly) should be mentioned. The authors analyzed massive datasets and provided clustering, but comparatively to the scale of the analysis, the fraction of novelties reported in the paper is relatively modest: the number of newly discovered larger protein families is comparatively small, and the authors highlight only a single new fold. Why is that? Is it because it will take years to carefully mine the datasets and figure out the remaining novelties, or has the worldwide structural biology community done a terrific job over the years of exploring the dark corners of the protein world to determine the majority of functional diversity? If the latter is true, it does NOT diminish the significance of this work and is NOT a negative result. To me, it would be a valuable positive message that could be highlighted.

4. Difficulty of functional annotation from structural similarity. This was “a trap” for structural genomics projects and something that more general scientists unfamiliar with the topic may miss. Having a structure (or good model) is not the same as having the function of the protein or domain. Functions are fluid in evolution; therefore, experimental exploration of structure-driven hypotheses is essential. The authors fully understand this and study one such family experimentally, but I think the general message along these lines could be strengthened.

5. Fonts in some figures are a bit too small, while there is empty room around to allow for the font size increase (e.g., Fig. 1). Sure, it is possible to zoom in on the PDF and see everything, but wouldn't it be more convenient for readers to have larger fonts when possible?

Author Rebuttals to Initial Comments:

Response to Referees

Referee #1:

A. Summary of the key results

This work presents a large-scale clustering of available protein sequences from UniProt50 (the UniProt database clustered to a maximum of 50% sequence identity) to identify uncharacterized families, i.e. groups of similar sequences that have not yet been functionally annotated. The uncharacterized families were defined as those that generally lack sequence-based annotations (domains, disorder, coiled coils, etc.) provided by the InterPro, UniProtKB, and UniParc databases, and were subsequently analyzed on a case-by-case basis, resulting in their better characterization.

B. Originality and significance

The authors present a solid bioinformatics study supported by experimental validation of some of the results (TumE-TumA toxin-anitotxin system, Fig. 5). While the paper is methodologically sound and takes advantage of all the recent advances in protein bioinformatics, I have some concerns about the study design, the main conclusions, and the availability of the data.

The authors point out (pp. 3-4, lines 119-121) that some of the uncharacterized families have corresponding high-quality AF2 models and some do not. Given this, it is unclear why the similarity network (Fig. 1) was computed only for UniRef50 clusters containing a high-quality model in the AFDB - first, the clustering is purely sequence-based (no structural information is required), second, as mentioned above, there are uncharacterized families for which the structural information is missing in the AFDB (this, however, does not mean that it was impossible to compute a model for them, but that it is missing in the AFDB).

This is a very good point, and we thank the reviewer for bringing to our attention that our motivations require further explanation. The main reason for the selection of this set was confidence in the underlying data and associated annotations. While we agree that there is a wealth of additional information in metagenomic data or in other data not covered by AFDB, we wanted to set up the basis for automated annotations using the new set of high confidence data now available to us thanks to deep learning approaches such as AlphaFold and ProtNLM. Thus, we selected UniProt sequences and specifically those where predicted structure information could be confidently leveraged at an unprecedented scale. We focus specifically on AlphaFold models with high confidence, as they are typically of a higher accuracy and reliability than those from ESMfold.

We have now made our motivations explicit throughout the main text. This includes:

(1) the summary paragraph, with the sentence *"In this study, we examine the extent to which the AlphaFold database has structurally illuminated this "dark matter" of the natural protein universe at high predicted accuracy."*,

(2) the start of the “*Functional darkness in UniProt and AFDB*” section, where the second sentence is now **“We focus our analysis on these as they have a higher confidence than those deposited in metagenomics databases such as MGnify.”**

(3) the end of the “*Functional darkness in UniProt and AFDB*” section, with the sentence **“Thus, there is a considerable proportion of proteins in UniProt that can not be automatically annotated, but that high confidence structural information can now be leveraged to gain insights about a substantial number of these”**,

(4) the start of the “*Sequence similarity network of AFDB90*” section, which now reads **“While UniRef50 provides groups of sequences that are overall similar at the sequence level, they do not reach the family and superfamily levels and do not account for local similarities. To reach these levels and put functionally dark clusters into evolutionary context, we constructed a large-scale sequence similarity network of all clusters where structural information can be confidently leveraged to support functional annotations.”**,

(5) The 4th paragraph in the “*Towards large-scale function annotation*” section, which now includes: **“ Furthermore, we focus only on proteins with high confidence predicted structures from AFDB, setting aside the wealth of potential darkness in metagenomic data for which structural models are also now available through the ESM Metagenomic Atlas”**

Additionally, while the clustering itself is sequence based, both our large-scale and in-depth analysis make use of additional information on top of the sequence network - specifically functional brightness, semantic diversity of names predicted by ProtNLM, and structural outlier scores when comparing AFDB models to the PDB. All of this additional information, along with the network, is what allowed us to find putative new families and folds - we now refer to the sequence similarity network as **“annotated sequence similarity network”** throughout the text and in all examples we now explicitly mention how this enriched network was used.

The first example (uncharacterized family/component 27, p. 5) further highlights the fact that the AF2 models were not really necessary for the annotation. An HHpred search with a representative sequence (AOA7X7MB17) clearly shows the homology to glycotransferases (STT3/PglB/AgIB) with very good e-values. In this case, structural searches were not essential (p. 5, lines 167-174) and could even be misleading, as I will try to explain below. The authors write: "This example illustrates the power of structural alignment to resolve undetected remote homology relationships". The structural search did identify other similar proteins (Fig. 4b), but I would be very cautious about claiming homology based on structural similarity alone. There are examples of structures that are similar, but the extent of their homology is debated (e.g. the class of "immunoglobulin-like"). With this in mind, I believe that Fig. 4a should include only sequence-based homologs. Otherwise, it suggests an equivalence of sequence and structural similarity in homology detection (which may be true in this case, but is not a general rule).

We have rephrased this example to better highlight how structural information was useful, and now also mention the sequence similarity found using HHpred. In this case, while structural information alone was not what led to the annotation described, inspecting the structural model is what highlighted a possible problem in the existing annotation and prompted further analysis. The sequence similarity network constructed specifically for these proteins and those found by structure searches substantiates the evolutionary relationship between the different families. We now write in this section:

“However, the predicted structural model superposes poorly to the YfhO family (TM-score 0.58, Fig. 3b), prompting a more in-depth investigation. (...) These results support the notion that component 27 belongs to the well-studied superfamily of transmembrane oligosaccharyl- and glycosyltransferases, but also indicate that it is a hitherto undescribed bacterial protein family. In this case, inspecting the AlphaFold model revealed possible inconsistencies in their automated annotation, illustrating the added value of structural models to guide sequence-based family classification.”

In addition, we added another example in the “*Semantic consistency across the network*” section that illustrates the value of combining both sequence and structure similarities to guide functional annotation and hypothesis generation. This corresponds to component 6’732, which has a high semantic diversity and although no clear global sequence similarity was found to proteins of known function, structural searches with the AFDB model highlighted a putative conserved catalytic site similar to that of EndoMS, a mismatch restriction endonuclease. The predicted fold was instrumental in putting forward a functional hypothesis based on an otherwise undetectable catalytic site conserved also at the sequence level.

In general, the conservation of rare or peculiar structural features, such as active/binding sites, along with topological and structural similarity, provide strong support for a common evolutionary origin, even in the absence of significant global sequence similarity. We hope that these additions, combined with the general restructuring of text throughout the manuscript, better tie these examples to the conclusions we wish to make: namely that a combination of data and approaches are often required for confident protein annotation and family curation, and that our enriched network significantly facilitates such efforts.

In contrast to the case study mentioned above, the analyses of “Structural outlier diversity in AFDB90” (p. 7 ff.) make use of the AFDB structural data, but do not seem to be related to the sequence-based clustering procedure of the AFDB90 subset. In other words, these studies, including the discovery of the beta-flower fold, could easily have been made into a separate paper, suggesting their rather loose connection to the rest of the work.

Thank you for highlighting that the relation between the structural outlier detection and the sequence-based clustering was not presented in a sufficiently clear manner. Just as we used the semantic diversity measure to prioritise from the wealth of dark connected components which ones to look closer into, the structural outlier score provided us with a different layer of information on top of the network. We have now rephrased and improved this section. We highlight how outliers distribute across the network, what their features generally are, and point out that many outliers are singletons. Given that a large number of proteins within our network lie in singleton nodes and thus methods such as the ones used in previous sections may not be informative enough, the beta-flower example demonstrates how the outlier score could be used to prioritise examples even within these singletons.

The relevant additions to the text in this section are:

“Just as semantic diversity revealed novelties in protein sequence space, we also investigated how different the predicted structural characteristics of proteins in our network are from the structures in the PDB. (...) While most fully dark and fully bright components do not contain structural outliers, the outlier content is significantly different between the two sets (Kolmogorov–Smirnov p -value = 5×10^{-81} , Extended data Fig. 4c). Fully dark components have on average a higher outlier content (21%) than fully bright (15%), but these only correspond to about half of the structural outliers. Indeed, 44% of outliers are singletons, i.e. UniRef50 clusters which do not form a component with at least 2 nodes, giving us a measure to prioritise even these cases for further analysis, as in the example below. (...) This, together with the different types of structural outliers described, highlights that the 3D context provided by the models in AFDB is highly informative for protein analysis efforts and that the structural space covered needs to be put into a coherent evolutionary, functional, and local structural context before any model, even with high predicted accuracy, is used as a reference.”

The above considerations lead to a main question about the main purpose of the work presented. Is it to provide starting points (i.e., uncharacterized families) for analyses such as those exemplified (pp. 5-9)? If this is the case, then first, as mentioned above, I trust that the analyses should not be limited to proteins that have counterparts in the AFDB, and second,

the data should be provided in a more readable form. Currently the only options are a huge 4GB file from Zenodo or the interactive AFDB90v4 atlas (see also point F).

The reviewer is correct, our goal is to highlight and facilitate the finding of interesting starting points that may correspond to unknown biological systems or spark new projects in molecular biology. To stress this, we now write:

(1) in the *"Towards large-scale function annotation"* section: ***"We demonstrate that this network is a rich source of putative novel protein folds, families and superfamilies, providing multiple starting points for further downstream studies. (...) When combined with traditional protein evolution approaches, structure-based comparisons, genomic context information, structure-based function prediction, and the conservation of local features such as active sites, we could gather support for common evolutionary origins, gain valuable insights into putative functions and put forward concrete testable hypotheses for experimental characterisation. (...) This information can now help guide further validation experiments, such as those carried out for TumE."***

(2) at the end of the *"Sequence similarity network of AFDB90"* section: ***"These fully dark components are fertile ground for novel family discovery, as exemplified by the two new families we describe below"***.

Our new title ***"Uncovering new families and folds in the natural protein universe"*** now also better reflects this point.

While we agree that much novelty still exists beyond AFDB, as elaborated above, we chose this specific dataset due to the confidence of the data that UniProt and AFDB provide, laying the basis for larger scale studies in the future, where even low confidence information could be included. Regarding the data access, we have made a number of changes both in the Zenodo repository and the Atlas web resource, as described in the reply to point F.

C. Data & methodology

The authors relied on state-of-the-art and well-established approaches to sequence and structure analysis. However, I would expect the data derived from metagenomes to be considered, as they may contain many "dark" families.

While we agree that metagenomic data is a rich source of darkness, UniProt is a higher confidence source of protein sequence and annotation data. Plus, AlphaFold models are typically of a higher accuracy and reliability than those from ESMfold. With this work we establish the basis for such an Atlas, and we plan in future efforts to expand to metagenomic data where we would then focus our research on the most effective way that confidence metrics can be used to enrich the network. We now close the *"Towards large-scale function annotation"* section with ***"We anticipate that further advances in deep learning-based methods for function prediction, remote homology detection and protein structure prediction will allow for analyses on an even larger scale, incorporating more diverse data sources with greater confidence."***

D. Appropriate use of statistics and treatment of uncertainties

Not applicable.

E. Conclusions: robustness, validity, reliability

Page 10, lines 401-403, "Our results indicate that 19% of this space is composed of protein families and superfamilies that are functionally dark and cannot, solely on the basis of sequence similarity, be annotated to a known protein family": The procedures for using structural similarity to detect homology are not fully established, and there are clearly cases where such similarity may not be the result of common ancestry. I think this should be addressed and discussed more directly in the paper, especially if the AFDB models remain the focus of the study.

We agree that any single source of information alone can not be used to confidently establish an evolutionary relationship, as already stated in the "*Towards large-scale function annotation*" section. In general, the conservation of rare structural features, such as active/binding sites, along with topological and structural similarity, provide strong support for a common evolutionary origin, even in the absence of significant global sequence similarity. This was the case for all our examples, including the beta-flower, which we now clarify and specifically mention in the same section: "***Indeed, the functional annotation of dark proteins, even from a purely computational perspective, requires a combination of data sources and approaches. It is crucial to combine individual predictions across connections in the network to increase the confidence of any hypothesis. Most of our examples had such support from both sequence and structure, and even for the novel β -flower fold, a singleton in our network, the presence of a semi-conserved sequence motif captured only due to local structural similarities allowed us to generate an initial classification.***"

We have also reiterated in all of our examples how sequence and structure information were employed to define new families, and specifically in the case of the beta-flower we made it clear how local sequence features highlighted by the structural alignment drove the classification of a clan that at sequence level would remain singletons, by including in the "*The β -flower fold*" section: "***Based on their global structural similarity and the presence of a semi-conserved [DNEQ]XXG sequence motif at the tip of the β -hairpin, and the repeat unit of both β -flowers and Tubby-like, the diversity of these proteins has been added to Pfam as the new entries PF21784, PF21785 and PF21786.***"

Page 10, lines 406-408, "Our findings highlight the power of large-scale similarity networks to curate protein function. In cases where simple homology transfer methods are not effective, pooling information from across the network can enhance remote homology detection": I am not sure which parts of the work the authors are referring to. The "large-scale similarity network" allows the detection of poorly annotated protein families, which can then be analyzed on a case-by-case basis, however I can't see how "pooling information from across the network can enhance remote homology detection". Authors are encouraged to be more specific in this regard.

With this point we were referring to combining the sequence similarity information brought by the network with the different layers of information annotated over its nodes, i.e. functional brightness, semantic diversity and structure outlier scores. To better highlight this in the main text, we have moved this comment to the "*Semantic consistency across the network*" section, which now states: "***Overall, pooling predictions across the network can help assess the consistency of automated annotation methods, especially in data-driven approaches.***" and clarified it in the "*Towards large-scale function annotation*" section: "***It is crucial to combine individual predictions across connections in the network to increase the confidence of any hypothesis.***"

F. Suggested improvements: experiments, data for possible revision

The atlas is visually appealing, but has limited functionality, and the 4GB flat file can only be used by experienced users. What I miss is an online resource that would allow easy searching (by sequence or structure) and browsing of the identified families (both well and poorly annotated). Such a resource could stimulate more analyses like those presented by the authors. Currently, I see no easy way to access the results of "the discovery of at least 281 putative new protein families" (p. 2, line 66), except those listed in Table S1.

We thank the reviewer for these suggestions as it has greatly increased the usability of our resource.

In order to facilitate the navigation, downloading and reusability of our data, we have now created a revised version of the Zenodo repository, where each raw data file associated with the network and its analysis can be downloaded individually. This repository also includes the CLANS maps for each of the examples mentioned in the text.

We have extended considerably the functionalities of the interactive Atlas. The additions are as follows:

- The contents and features of individual communities are now displayed in tabular form, which is interactive and communicates with the network itself (clicking on a row highlights the corresponding node and vice-versa).
- All annotation data can now be downloaded not only for individual communities but also for components, and for any user query in both tabular (.csv and .tsv) and .json formats.
- The titles of the individual proteins now include the updates introduced by UniProt, so that now they are also queryable.
- We have added sequence and structure-based searches, which can be carried out both by uploading an input fasta or PDB file, or pasting the target sequence or PDB-formatted atomic coordinates into the search box. If a sequence is provided, it searches for similar sequences in the AlphaFold database, and if a structure is provided, it uses the FoldSeek API to search for structural matches in the AlphaFold UniProt50 database. The matches are then mapped to their corresponding communities, which are then displayed in a separate zoom box that communicates with the main network. Matched communities are coloured based on the score of the match, and the results from all searches can also be downloaded.
- Semantic diversity scores have been added as an additional layer on the network.
- A help page (found by clicking the "About AFDBv490" button) was added to the resource summarising all these features, how to use them and how to interpret them.

Regarding the 281 (now 290) new putative families found, we now list them all in supplementary table S2, including those that were not added yet to Pfam, and semantic diversity scores have been added as an additional layer on the network. With this, it is possible now to query each individual connected component in the network or find them based on their diversity score, and to download any associated metadata easily.

G. References

The bibliography seems adequate. However, the lack of reference to ESMFold and ESM Atlas is very surprising. The ESM Atlas covers much more sequence space than the AFDB, and perhaps there is more "darkness" in it?

We agree that metagenomic data likely contains a larger amount of “darkness” and we cited the ESM Atlas in the “Towards large-scale function annotation” section when referring to the current large-scale efforts to analyse the diversity of the protein universe, but it was not explicitly mentioned by name. We rectified this now when commenting on prospective research efforts in this section by adding the following sentence: **“Furthermore, we focus only on proteins with high confidence predicted structures from AFDB, setting aside the wealth of potential darkness in metagenomic data for which structural models are also now available through the ESM Metagenomic Atlas ”**

H. Clarity and context

The paper, including the abstract, is well written and clear. In general, however, it may be misleading in suggesting direct use of the AF2 models: “we measured the extent to which AlphaFold has illuminated the <dark matter> of the natural protein universe, and modelled the diversity of the proteins it covers”. As I discussed above, the AFDB may be helpful/essential in analyzing some of the identified uncharacterized families, but its entire content (>200 million structures) is not really the focus of the work.

We have rephrased this sentence in the abstract: **“In this study, we examine the extent to which the AlphaFold database has structurally illuminated this “dark matter” of the natural protein universe at high predicted accuracy”** and in the Introduction: **“We revised their proportion, evaluated how many of them now have high confidence structural models that can be leveraged for downstream analysis, and constructed for the first time an annotated and interactive sequence similarity network with millions of proteins.”** and further reiterate the reasons behind the subset of AFDB analysed throughout the text, as described in the previous answers.

In its current form, the paper promises a combined structure-sequence study of protein “dark matter”, but in fact provides a sequence-based pipeline for identifying unannotated (“dark”) protein families. Such a pipeline is not very novel, given the availability of methods such as MMseqs2 and DIAMOND, and could be considered a simple pre-screening step for each of the case studies presented (regardless of whether it ultimately requires the use of structural predictions, as in the case of beta-flower, or not, as in the case of glycotransferases). This brings up another important point. While each of the case studies is interesting and well presented, they do not form a coherent story that supports the main promises of the paper (using the whole AFDB to uncover protein “dark matter”), leaving the impression that the submitted work is a compilation of a few good but independent projects.

We thank the reviewer for this comment as it helped us rephrase multiple sections of the manuscript to better reflect the message that we wished to convey, namely that combining multiple sources of data at large scale allows us to generate an integrative birds-eye-view of the catalogued protein universe which facilitates the discovery, annotation and curation of protein families across sequence space. While indeed the methods and approaches employed are not novel, the novelty of our work lies in their combination and scale. We have constructed the largest annotated and interactive protein sequence similarity network as a top-down view of the (currently high confidence only) protein universe, that contains much more information than the individual databases (UniProt, AlphaFold) and methods (MMSeqs, ProtNLM, etc.) provide. While a network representation of the protein universe is not new in the field, which we now also cite in the “Towards large-scale function annotation” section, it was never constructed at this scale.

This is a timely effort thanks to the explosion of predicted data now available, as we now state in the Introduction: **“Notwithstanding, deep learning based approaches have recently achieved unprecedented accuracy, with AlphaFold2 at the forefront. Its success drove the establishment of the AlphaFold database (AFDB), which contains predicted structural models for about 215 million natural protein sequences from UniProt, including many of the unannotated proteins. At the**

same time, deep learning-based approaches have also recently been employed for predicting functional properties from structure and protein names from sequence.” and allows for large-scale detection of undetected evolutionary links, as now stated in the “Towards large-scale function annotation” section: *“Remarkably, 40% of these dark clusters connect to bright UniRef50 clusters, revealing potential evolutionary relationships for over 700’000 unique proteins.”* but highlights multiple regions in this high confidence protein space that may correspond to novel protein (super)families or folds, as we described in previous answers and throughout the text now.

As computational prediction methods using complementary data sources further progress, we aim to add these insights to our network and thus provide a continuously updated resource with state-of-the-art information and predictions at the fingertips of life scientists.

Referee #2:

We are entering the era of comprehensive coverage of protein sequence-structure-function space due to revolutionary progress in protein structure prediction. This very timely and exciting manuscript capitalizes on these developments. It offers a comprehensive clustering of ~200 million protein structure models in the AF database and skims through the highlights of the findings.

This work is significant because most researchers do not have the computational capabilities to access the entire AF database and explore it for protein families of their interest. This work equips them with an easier way to access the data. In my opinion, this is the most significant value of this work for the research community.

Furthermore, the paper stands on its own, making a unique contribution to evolutionary protein classification and structure-function relationship. The distribution of protein clusters and functional annotations among them, as well as other general statistics reported in the paper, are illuminating and instructive. Biological discoveries (one is validated experimentally) are interesting. For all these reasons, I believe this paper will be noted by a broad audience and widely cited.

The methods used are solid and are described in detail so that they can be reproduced.

We thank the reviewer for their positive comments and agree that such work and resources can be of great use to the broader life sciences community.

The following improvements would further increase the impact of this work.

1. Access to the data and clusters: Many biologists with developing computational skills would like to access and explore protein clusters obtained in this work. Therefore, it would be helpful to explain more about what is offered for such exploration and how it should be done. I see that all the data are offered for download, but it takes some computational

expertise (really!) to figure out what is what. Simplifying this process (e.g., answering questions such as: how to find which cluster my favorite protein belongs to and what functional annotations are associated with this cluster) would help. To be clear, I am not asking for much work here, but for some supplementary file (kind of a simplified and easily accessible “readme”) that explains what can be downloaded and how and answers some possible FAQs about these datasets.

We thank the reviewer for this suggestion. In order to facilitate the navigation, downloading and reusability of our data, we have now created a revised version of the Zenodo repository, where each raw data file associated with the network and its analysis can be downloaded individually. This repository also includes the CLANS maps for each of the examples mentioned in the text. Each file is now described in the Zenodo description, and is also explained in the corresponding documented jupyter notebooks available through the dedicated Github repository.

Following the comments from reviewer 1, we have also extended considerably the functionalities of the interactive Atlas, which allows for a better usage of our data. The additions are as follows, and include now a README page that guides users on how use the data:

- The contents and features of individual communities are now displayed in tabular form, which is interactive and communicates with the network itself (clicking on a row highlights the corresponding node and vice-versa).
- All annotation data can now be downloaded not only for individual communities but also for components, and for any user query in both tabular (.csv and .tsv) and .json formats.
- The titles of the individual proteins now include the updates introduced by UniProt, so that now they are also queryable.
- We have added sequence and structure-based searches, which can be carried out both by uploading an input fasta or PDB file, or pasting the target sequence or PDB-formatted atomic coordinates into the search box. If a sequence is provided, it searches for similar sequences in the AlphaFold database, and if a structure is provided, it uses the FoldSeek API to search for structural matches in the AlphaFold UniProt50 database. The matches are then mapped to their corresponding communities, which are then displayed in a separate zoom box that communicates with the main network. Matched communities are coloured based on the score of the match, and the results from all searches can also be downloaded.
- Semantic diversity scores have been added as an additional layer on the network.
- A help page (found by clicking the “About AFDv490” button) was added to the resource summarising all these features, how to use them and how to interpret them.

2. Some sections in the results are riddled with numbers and read like free-text tables. Is it necessary? I personally was a bit bored reading these sections. Can they be rephrased somewhat and made more interesting, with these texts transferred to the supplement? I’m not insisting on it, it is just my opinion, and I understand that the authors put much work into getting these numbers, and they must be proud of what they got.

We have now shortened and simplified the main text to keep only the most relevant numbers, and moved some of the remaining statistics to figure legends and methods. We are grateful for this suggestion as it has significantly improved readability.

3. A comment which should not detract from the value of this study but seems like an interesting result that (possibly) should be mentioned. The authors analyzed massive datasets and provided clustering, but comparatively to the scale of the analysis, the fraction of novelties reported in the paper is relatively modest: the number of newly discovered larger protein families is comparatively small, and the authors highlight only a single new fold. Why is that? Is it because it will take years to carefully mine the datasets and figure out the remaining novelties, or has the worldwide structural biology community done a terrific job over the years of exploring the dark corners of the protein world to determine the majority of functional diversity? If the

latter is true, it does NOT diminish the significance of this work and is NOT a negative result. To me, it would be a valuable positive message that could be highlighted.

One reason for the relatively “low” fraction of novelty is due to the fact that we focus on proteins with high confidence structural models in AFDB. Indeed, when checking the proportion of dark UniRef50 clusters across UniProt and AFDB at different pLDDT levels, there was a consistent decrease with an increase in predicted pLDDT cutoffs (Extended data figure 1). Nonetheless, our chosen subset still holds a significant quantity of proteins that appeared “dark” according to UniProt and InterPro annotations, which, as we can now confirm through the network, are more likely to be divergent forms of functionally characterised proteins. While this doesn't yield a specific functional indication, it aids in their annotation and classification.

In the remaining fraction, however, there seems to be a sizable presence of novelty, and we've looked into various methods of highlighting these (component brightness, semantic diversity, and structure outlier scores). But actually going from such highlighted potential novelties to in-depth characterisation, especially the required experimental characterization, is a task that the reviewer rightly points out will take years. While we cannot hope to accomplish such large-scale in-depth experimental characterization ourselves, we believe our work and resource contribute by spotlighting these cases and assisting life scientists in determining which ones deserve further investigation. As it has already done, it also significantly facilitates protein curation efforts by biocurators.

We have addressed these messages:

- in the “*Sequence similarity network of AFDB90*” section: ***“Remarkably, 40% of these connect to bright UniRef50 clusters, revealing previously undetected potential evolutionary relationships for over 700’000 unique proteins.”***
- in the “*Towards large-scale function annotation*” section: ***“We demonstrate that this network is a rich source of putative novel protein folds, families and superfamilies, providing multiple starting points for further downstream studies. (...) We find that many functionally unannotated proteins are remote homologs of annotated ones, relationships which can now be easily explored. Additionally, over 1 million proteins belong to completely unannotated connected components, many of which cannot be named consistently using the most recent deep-learning-based approaches or contain proteins with structural features distinct from what is seen in the PDB. (...) Though we could already highlight a significant proportion of novelty, in-depth exploration combining multiple sources of evidence could only be carried out for a small number of families and folds. Thus, the examples we discuss are the low-hanging fruit of uncharacterised or unannotated protein families, and they are only the tip of the iceberg.”***

4. Difficulty of functional annotation from structural similarity. This was “a trap” for structural genomics projects and something that more general scientists unfamiliar with the topic may miss. Having a structure (or good model) is not the same as having the function of the protein or domain. Functions are fluid in evolution; therefore, experimental exploration of structure-driven hypotheses is essential. The authors fully understand this and study one such family experimentally, but I think the general message along these lines could be strengthened.

This is a very good point, also mentioned by reviewer 1, and we agree. Structural data is only one piece of the puzzle, which AFDB now tries to provide, but with limitations. These data are useful in generating and prioritising hypotheses, but they are not a replacement for experiments. We now comment on this specifically in the “*Towards large-scale function annotation*” section, where we write ***“When combined with traditional protein evolution approaches, structure-based comparisons, genomic context information, structure-based function prediction, and the conservation of local features such as active sites, we could gather support for common evolutionary origins, gain valuable insights into putative functions and put forward concrete testable hypotheses for experimental characterisation. Indeed, the functional annotation of dark proteins, even from a purely computational perspective, requires a combination of data sources and approaches. It is crucial to combine individual predictions across connections in the network to increase the confidence of any hypothesis. Most of our examples had such support from both sequence and structure, and even for the novel β -flower fold, a singleton***

in our network, the presence of a semi-conserved sequence motif captured only due to local structural similarities allowed us to generate an initial classification. This information can now help guide further validation experiments, such as those carried out for Tume.”

We have also reiterated in all of our examples how sequence and structure information were employed to define new families, and specifically in the case of the beta-flower we made it clear how local sequence features highlighted by the structural alignment drove the classification of a clan that at sequence level would remain singletons. ***“Based on their global structural similarity and the presence of a semi-conserved [DNEQ]XXG sequence motif at the tip of the β -hairpin, and the repeat unit of both β -flowers and Tubby-like, the diversity of these proteins has been added to Pfam as the new entries PF21784, PF21785 and PF21786, which together with the Tubby C-terminal domain now form the CL0395 clan.”***

5. Fonts in some figures are a bit too small, while there is empty room around to allow for the font size increase (e.g., Fig. 1). Sure, it is possible to zoom in on the PDF and see everything, but wouldn't it be more convenient for readers to have larger fonts when possible?

We fixed all the fonts and simplified some figures, especially Fig. 1, so that it is easier to read and interpret.

Reviewer Reports on the First Revision:

Referees' comments:

Referee #1:

All the points have been addressed by the authors' responses and changes to the text. The improvement in data availability is also much appreciated.

Referee #2:

The authors carried out a very careful and detailed revision of the manuscript and they addressed all comments quite well. I think that the manuscript is much improved after revision and is easier to follow. I did not find any major issues, even after carefully studying the manuscript.

Some further edits for clarity can be made throughout, e.g., "most catalogued natural proteins" in the summary is confusing. Is it "nearly all proteins that are known" or "extensively catalogued proteins"? More naïve readers may not understand this. The word "catalogued" may not be best choice here because it is not clear what it means due to multiple meanings. "Pfam" probably should be "Pfam database."

Overall, I think this paper represents an impressive achievement.